# Characterization of nighttime formation of particulate organic nitrates based on high-resolution aerosol mass spectrometry in an urban atmosphere in China

Kuangyou Yu[1,2,*], Qiao Zhu[1,*], Ke Du[2], Xiao-Feng Huang[1]

[1]Key Laboratory for Urban Habitat Environmental Science and Technology, School of Environment and Energy, Peking University Shenzhen Graduate School, Shenzhen, 518055, China.

[2]Department of Mechanical and Manufacturing Engineering, University of Calgary, Calgary, Canada.

* Authors have equal contribution.

**Abstract.** Organic nitrates are important atmospheric species that significantly affect the cycling of NOx and ozone production. However, characterization of particulate organic nitrates and their sources in polluted atmosphere is a big challenge and has not been comprehensively studied in Asia. In this study, an Aerodyne high-resolution time-of-flight aerosol mass spectrometer (HR-ToF-AMS) was deployed at an urban site in China from 2015 to 2016 to characterize particulate organic nitrates in total nitrates with high time resolution. Based on the cross validation of two different data processing methods, organic nitrates were effectively quantified to contribute a notable fraction of organic aerosol (OA): 9-21% in spring, 11-25% in summer, 9-20% in autumn, while very small fraction in winter. The good correlation between organic nitrates and fresh secondary organic aerosol (SOA) at night, as well as the diurnal trend of size distribution of organic nitrates, indicated a key role of nighttime local secondary formation of organic nitrates in Shenzhen. Furthermore, theoretical calculations of nighttime SOA production of $NO_3$ reactions with volatile organic compounds (VOCs) measured during the spring campaign were performed, resulting in three biogenic VOCs (α-pinene, limonene, and camphene) and one anthropogenic VOC (styrene) identified as the possible key VOC precursors for particulate organic nitrates. The comparison with similar studies in the literature implied that nighttime particulate organic nitrates formation is highly relevant with NOx levels. This study proposes that unlike the documented cases in the United States and Europe, modeling nighttime particulate organic nitrate formation in China should incorporate not only biogenic VOCs but also anthropogenic VOCs for urban air pollution, which needs the support of relevant smog chamber studies in the future.

*Correspondence to*: X.-F. Huang (huangxf@pku.edu.cn)

## 1. Introduction

Organic nitrates (ON) in aerosols have an important impact on the fate of $NO_X$ and ozone production (Lelieveld et al., 2016), which can be formed in a minor channel of the reaction between peroxy radicals and NO (R1 and R2) (usually, an increased fraction of this reaction leads to the formation of alkoxy radicals and $NO_2$ (R3)) or via the $NO_3$-induced oxidation of unsaturated hydrocarbons (R4). Even though some recent studies have suggested that the formation of organic nitrates from peroxy radicals and NO may play a larger role than previously recognized (Teng et al., 2015, 2017), yields of organic nitrates via $NO_3$ reacting with alkenes are generally much higher (Fry et al., 2009; Ayres et al., 2015; Boyd et al.,2015,2017).

$$RH + OH + O_2 \rightarrow RO_2 + H_2O \quad (R1)$$

$$RO_2 + NO \rightarrow RONO_2 \quad (R2)$$

$$RO_2 + NO \rightarrow RO + NO_2 \quad (R3)$$

$$R = R' + NO_3 \rightarrow R(ONO_2)\ R' \quad (R4)$$

Several methods have been developed to directly measure total organic nitrates (gas + particle) in the real atmosphere. For example, Rollins et al.(2012) used a thermal-dissociation laser-induced fluorescence technique (TD-LIF) to observe organic nitrates in the United States; Sobanski et al. (2017) measured organic nitrates in Germany using the thermal dissociation cavity ring-down spectroscopy (TD-CRDS). Field and laboratory studies around the world indicated that particulate organic nitrates could contribute a large portion of secondary organic aerosol (SOA) (Rollins et al., 2012; Xu et al., 2015a; Fry et al., 2013; Ayres et al., 2015; Boyd et al., 2015; Lee et al., 2016). Recently, researchers have proposed some estimation methods for particle-phase organic nitrates based on aerosol mass spectrometry (AMS) with high time resolution (Farmer et al., 2010; Hao et al., 2014; Xu et al., 2015a, 2015b). Ng et al. (2017) reviewed the nitrate radical chemistry and the abundance of particulate organic nitrates in the United States and Europe, and further concluded that particulate organic nitrates are formed substantially via $NO_3$+BVOC chemistry, which plays an important role in SOA formation. Unfortunately, relevant Chinese datasets are scarce yet and not included in this review. This was because (1) the contributions of organic nitrates in SOA and total nitrates in Chinese atmosphere remain poorly understood; (2) the anthropogenic and biogenic precursor emissions in China are significantly different from those in the United States and Europe, and thus cannot be easily estimated. To our best knowledge, few studies have investigated the concentrations and formation pathways of particulate organic nitrates in China. Xu et al. (2017) estimated the mass concentration of organic nitrogen in Beijing using AMS, but in this study they ignored the contribution of $NO_X^+$ family, which are the major fragments of organic nitrates.

Shenzhen is a megacity of China in a subtropical region, where NOx involved photochemical reactions are very active, given considerable biogenic and anthropogenic VOC emissions (Zhang et al., 2008). To assess the evolution of particulate organic nitrates in a polluted urban atmosphere, we deployed an Aerodyne high-resolution time-of-flight aerosol mass spectrometer (HR-ToF-AMS) and other instruments in Shenzhen from 2015 to 2016 in this study. Organic nitrates and their contributions to OA in different seasons were estimated by different methods using the HR-ToF-AMS datasets, based on which, the secondary formation pathway of particulate organic nitrates in Shenzhen was further explored.

## 2. Experiment methods

### 2.1 Sampling site and period

The sampling site (22.6°N, 113.9°E) was on the roof (20 m above ground) of an academic building on the campus of Peking University Shenzhen Graduate School (PKUSZ), which is located in the western urban area in Shenzhen (Figure 1). This site is mostly surrounded by subtropical plants without significant anthropogenic emission sources nearby, except a local road that is ~100 m from the site. In this study, we used the statistical data from the Meteorological Bureau of Shenzhen Municipality (http://www.szmb.gov.cn/site/szmb/Esztq/index.html) as the reference data to determine the sampling periods for four different seasons during 2015-2016, as shown in Table 1.

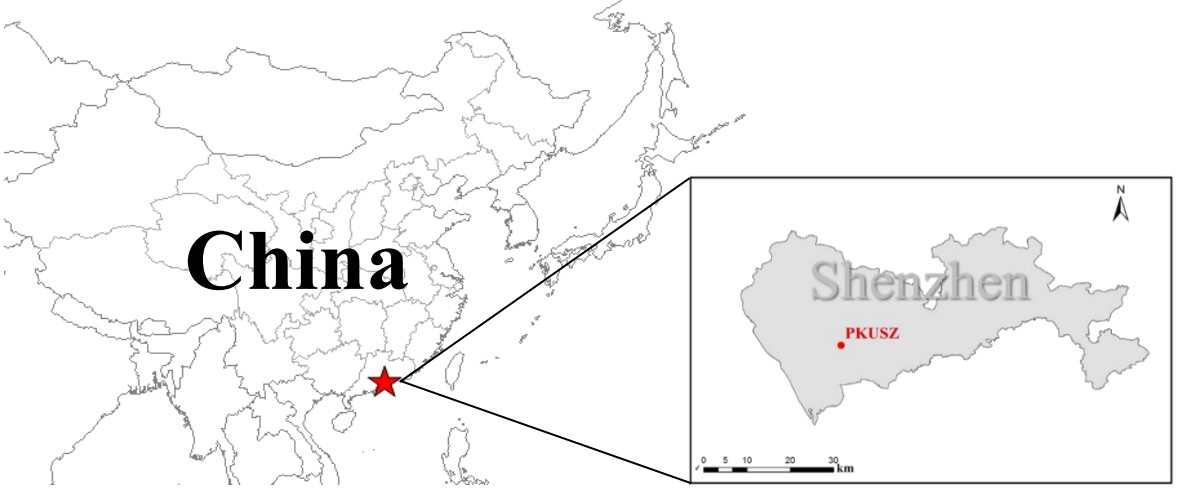

**Figure 1.** The location of the sampling site.

**Table 1.** Meteorological conditions, $PM_1$ species concentrations and relevant parameters for different sampling periods in Shenzhen.

| | Sampling period | 4.1-4.30, 2016 | 8.1-8.31, 2015 | 11.4-11.30, 2015 | 1.21-2.3, 2016 |
| --- | --- | --- | --- | --- | --- |
| | | **Spring** | **Summer** | **Autumn** | **Winter** |
| **Meteorology** | T (°C) | 24.5±2.5 | 29.0±3.0 | 23.6±3.7 | 10.7±4.7 |
| | RH (%) | 78.0±12.7 | 71.2±17.5 | 68.2±15.8 | 75.4±18.7 |
| | WS (m s$^{-1}$) | 1.4±0.8 | 1.0±0.7 | 1.2±0.7 | 1.5±0.8 |
| **Species** | Org | 4.3±3.2 | 10.0±6.9 | 7.8±5.9 | 5.1±3.5 |
| | $SO_4^{2-}$ | 3.2±1.8 | 5.8±3.3 | 2.3±1.5 | 1.9±1.2 |
| | Total $NO_3^-$ | 0.96±1.4 | 0.91±0.90 | 1.3±1.4 | 1.6±1.0 |

| (μg m$^{-3}$) | $NH_4^+$ | 1.4±0.8 | 2.0±1.1 | 1.1±0.8 | 1.2±0.6 |
|---|---|---|---|---|---|
| | $Cl^-$ | 0.14±0.19 | 0.03±0.05 | 0.22±0.36 | 0.64±0.85 |
| | BC | 1.9±2.1 | 2.4±1.6 | 3.5±2.6 | 2.4±1.5 |
| | Total | 12.0±8.9 | 15.1±13.8 | 11.8±9.5 | 12.2±7.2 |
| ON relevant parameters | $R_{NH4NO3}$ | 2.80 | 3.20 | 3.32 | 3.48 |
| | $R_{obs}$ | 3.74 | 6.14 | 4.30 | 3.55 |
| | Fraction of positive numbers of $R_{obs}$- $R_{NH4NO3}$ | 99% | 99% | 84% | 47% |

## 2.2 Instrumentation

### 2.2.1 High Resolution Time-of-Flight Aerosol Mass Spectrometer

During the sampling periods, chemical composition of non-refractory PM$_1$ was measured by an Aerodyne HR-ToF-AMS, and detailed descriptions of this instrument are given in the literature (DeCarlo et al., 2006; Canagaratna et al., 2007). The setup and operation of the HR-ToF-AMS can be found in our previous publications (Huang et al., 2010, 2012; Zhu et al.,2016). To remove coarse particles, a PM$_{2.5}$ cyclone inlet was installed before the sampling copper tube with a flow rate of 10 l min$^{-1}$. Before entering the AMS, the sampled air was dried by a nafion dryer (MD-070-12S-4, Perma Pure Inc.) to eliminate the potential influence of relative humidity on particle collection (Matthew et al., 2008). The ionization efficiency (IE) calibrations were performed using pure ammonium nitrate every two weeks. The relative ionization efficiencies (RIEs) used in this study were 1.2 for sulfate, 1.1 for nitrate, 1.3 for chloride, 1.4 for organics and 4.0 for ammonium, respectively (Jimenez et al., 2003). Composition-dependent collection efficiencies (CEs) were applied to the data according to the method in Middlebrook et al. (2012). The instrument was operated at two ion optical modes with a cycle of 4 min, including 2 min for the mass-sensitive V-mode and 2 min for the high mass resolution W-mode. The HR-ToF-AMS data analysis was performed using the software SQUIRREL (version 1.57) and PIKA (version 1.16) written in Igor Pro 6.37 (Wave Metrics Inc.) (http://cires1.colorado.edu/jimenezgroup/ToFAMSResources/ToFSoftware / index.html).

### 2.2.2 Other co-located instruments

In addition to the HR-ToF-AMS, other relevant instruments were deployed at the same sampling site. An aethalometer (AE-31, Magee) was used for measurement of refractory black carbon (BC) with a resolution of 5 min. An SMPS system (3775 CPC and 3080 DMA, TSI Inc.) was used to obtain the particle number size distribution in 15–615 nm (mobility diameter) with a time resolution of 5 min. Ozone and NO$_X$ were measured by a 49i ozone analyzer and a 42i nitrogen oxide analyzer (Thermo Scientific), respectively. In the spring campaign, ambient VOC concentrations were also measured using an on-line VOC monitoring system (TH-300B, Tianhong Corp.), including an ultralow-temperature preconcentration cold trap and an

automated in-situ gas chromatograph (Agilent 7820A) equipped with a mass spectrometer (Agilent 5977E). The system had both a flame ionization detector (FID) gas channel for C2–C5 hydrocarbons and a mass spectrometer (MS) gas channel for C5–C12 hydrocarbons, halohydrocarbons and oxygenated VOCs. A complete working cycle of the system was one hour and included six steps: sample collection, freeze-trapping, thermal desorption, GC-FID/MS analysis, heating and anti-blowing purification. The sample collection time was 5 min. The sampling flow speed was 60 ml min$^{-1}$. The anti-blowing flow speed was 200 ml min$^{-1}$. The calibration of over 100 VOCs was performed using mixed standard gas before and after the campaign. Detection limits for most compounds were near 5 pptv. More description of this instrument can be found in Wang et al. (2014).

**2.3 Organic nitrates estimation methods**

In this study, we used two independent methods to estimate particulate organic nitrates based on the AMS data, following the approaches in Xu et al. (2015b). The first method is based on the $NO^+/NO_2^+$ ratio ($NO_X^+$ ratio) in the HR-AMS spectrum. Due to the very different $NO_X^+$ ratios of organic nitrates and inorganic nitrate (i.e., $R_{ON}$ and $R_{NH4NO3}$, respectively) (Farmer et al., 2010; Boyd et al., 2015; Fry et al., 2008; Bruns et al., 2010), the $NO_2^+$ and $NO^+$ concentrations of organic nitrates ($NO_{2,ON}$ and $NO_{ON}$) can be quantified with the HR-AMS data via Eqs. (1) and (2), respectively (Farmer et al., 2010):

$$NO_2^+{}_{,ON} = \frac{NO_2^+{}_{,obs} \times (R_{obs} - R_{NH4NO3})}{R_{ON} - R_{NH4NO3}} \quad (1)$$

$$NO^+{}_{ON} = R_{ON} \times NO_{2,ON} \quad (2)$$

where $R_{obs}$ is the $NO_X^+$ ratio from the observation. The value of $R_{ON}$ is difficult to determine because it varies between instruments and precursor VOCs. However, $R_{NH4NO3}$ was determined by IE calibration using pure NH4NO3 every two weeks for each campaign and the results showed stable values: In spring, the average $R_{NH4NO3}$ was 2.66 for the first IE calibration and 2.94 for the second one; in summer, the average $R_{NH4NO}$ was 3.05 and 3.34 for the first and second IE calibrations, respectively; in autumn, the average $R_{NH4NO3}$ was 3.33 and 3.31 for the first and second IE calibrations, respectively; in winter, the average $R_{NH4NO3}$ was 3.45 and 3.51 for the first and second IE calibrations, respectively. We adopted the $R_{ON}/R_{NH4NO3}$ estimation range (from 2.08 to 3.99) for variation of precursor VOCs in the literature to determine $R_{ON}$ (Farmer et al., 2010; Boyd et al., 2015; Bruns et al., 2010; Sato et al., 2010; Xu et al., 2015b), and thus two $R_{ON}$ values were calculated for each season to provide the upper bound (NO3_org_ratio_1) and lower bound (NO3_org_ratio_2) of NO3,org mass concentration.

The second method is based on the traditional positive matrix factorization (PMF) analysis of HR organic mass spectra, which resolves different organic factors (Zhang et al., 2011; Ng et al., 2010; Huang et al., 2013). Combined with $NO^+$ and $NO_2^+$ ions, the same analysis of HR organic mass spectra was performed to separate $NO^+$ and $NO_2^+$ ions into different organic factors and an inorganic nitrate factor (Hao et al., 2014; Xu et al., 2015b). The PMF analysis procedures in this study can be found in our previous publications (Huang et al., 2010; Zhu et al., 2016; He et al., 2011), resulting in three organic factors and one inorganic factor in spring, summer and autumn: a hydrocarbon-like OA (HOA) characterized by $C_nH_{2n+1}^+$ and $C_nH_{2n-1}^+$ and O/C of 0.11 to 0.18, a less-oxidized oxygenated OA (LO-OOA) characterized by $C_xH_yO_z^+$ especially $C_2H_3O^+$ and O/C of 0.28 to 0.70, a more-oxidized oxygenated OA (MO-OOA) also characterized by $C_xH_yO_z^+$ especially $CO_2^+$ and O/C of 0.78 to 1.24, and a

nitrate inorganic aerosol (NIA) characterized by overwhelming $NO^+$ and $NO_2^+$, as indicated in Fig S6. According to the
diagnostic plots of the PMF analysis shown in Figure S2 to S4, the same organic factors as those in the traditional PMF analysis
of only organic mass spectra were obtained. The $NO^+$ and $NO_2^+$ ions were distributed among different OA factors and the NIA
factor; thus the concentrations of nitrate functionality ($NO_{org}^+$ and $NO_{2,org}^+$) in organic nitrates ($NO_{3,\ org}$) are equal to the sum of
$NO_2^+$ and $NO^+$ via Eqs. (3) and (4), respectively (Xu et al., 2015b):
$$NO_{2,org}^+ = \sum([OA\ factor]_i \times f_{NO2,i}) \quad (3)$$
$$NO_{org}^+ = \sum([OA\ factor]_i \times f_{NO,i}) \quad (4)$$
where [$OA\ factor$]$_i$ represents the mass concentration of OA factor $i$, and $f_{NO2,i}$ and $f_{NO,i}$ represent the mass fractions of $NO_2^+$
and $NO^+$, respectively.
It should be noted that the 4-factor solution seemed to have a "mixed factor" problem to some extent (Zhu et al., 2018). For
example, HOA mixed with COA (clear $C_3H_3O^+$ in m/z 55 for spring, summer and autumn) (Mohr et al., 2012), and BBOA
mixed with LO-OOA (clear m/z 60 and 73 signals in LO-OOA in autumn) (Cubison et al., 2011). However, running PMF with
more factors would produce unexplained factors but less influence on the apportionment of $NO^+$ and $NO_2^+$ ions between
organic nitrates and inorganic nitrate (Table S1). In addition, the standard deviations of $NO^+$ and $NO_2^+$ ions in the OA factors
across different FPEAK values (from −1.0 to 1.0) were very small (Table S2). Therefore, the 4-factor solution was used for
quantifying organic nitrates in spring, summer and autumn.

## 3. Results and discussion

### 3.1 Organic nitrates estimation

Table 2 shows the concentrations of nitrate functionality in organic nitrates (i.e., $NO_{3,\ org}$), estimated by both the $NO^+/NO_2^+$
ratio method and PMF method, as well as their contributions to the total measured nitrate. It should be noted that the small
difference between the average $R_{obs}$ and $R_{NH4NO3}$ in winter leads to a large portion of negative data using the $NO^+/NO_2^+$ ratio
method (Table 1). The result from the PMF method shows that the contribution of organic nitrates in total nitrates is only 4.2%
in winter (Figure S6), suggesting a negligible contribution of organic nitrates. Thus, we will only discuss organic nitrate
estimation results in spring, summer and autumn. The analytical outcomes by the $NO^+/NO_2^+$ ratio method and by the PMF
method consistently suggest that organic nitrates had the highest ambient concentration (0.34-0.53 µg m$^{-3}$) and proportion in
total nitrates (41-64%) in summer among the different seasons. This finding agrees with the finding in (Ng et al., 2017) and it
implies a seasonal trend in comparison with that of total nitrates in Table 1. Assuming the average molecular weight of organic
nitrates of 200 to 300 g mol$^{-1}$ (Rollins et al., 2012), we found that organic nitrates contributed 9-21% to OA in spring, 11-25%
in summer, and 9-20% in autumn.
In the PMF method, the mass fractions of organic nitrates in HOA, LO-OOA and MO-OOA were 31%, 49% and 20%,
respectively, in spring; 28%, 52% and 20%, respectively, in summer; 30%, 46% and 24%, respectively, in autumn. The major
fraction of organic nitrates occurring in LO-OOA for the three seasons implied that organic nitrates were mostly related to
fresher secondary OA formation. The NIA factors in all seasons were dominated by but are not limited to $NO^+$ and $NO_2^+$.
Some organic fragments, such as $CO_2^+$ and $C_2H_3O^+$, are also part of these factors, which agreed with the findings in the
literature (Hao et al., 2014; Xu et al., 2015b; Sun et al., 2012). This indicated the potential interference of organics in the NIA
factor. It is also worth to be noted that the $NO^+/NO_2^+$ ratios in NIA (2.93 for spring, 3.53 for summer and 3.54 for autumn)
were higher than that for pure $NH_4NO_3$ (Table 1), indicating an underestimation of $NO_{3.org}$ concentration by the PMF method.
This finding may also explain the reason that the concentration of $NO_{3.org}$ estimated using the PMF method was always close
to the lower estimation bound of $NO_{3.org}$ concentration estimated using the $NO^+/NO_2^+$ ratio method in each season (Table 2).
**Table 2.** Summary of organic nitrates estimations using the $NO^+/NO_2^+$ ratio method and the PMF method

| Sampling period | $NO^+/NO_2^+$ ratio method | | | | PMF method | |
|---|---|---|---|---|---|---|
| | $NO_{3,org}$ ($\mu g\ m^{-3}$)[a] | | $NO_{3,org}/NO_3$ | | $NO_{3,org}$ ($\mu g\ m^{-3}$)[b] | $NO_{3,org}/NO_3$ |
| | lower | upper | lower | upper | | |
| **Spring** | 0.12 | 0.19 | 13% | 21% | 0.12 | 12% |
| **Summer** | 0.34 | 0.53 | 41% | 64% | 0.39 | 43% |
| **Autumn** | 0.21 | 0.33 | 16% | 25% | 0.21 | 16% |
| **Winter** | / | / | / | / | 0.07 | 4.2% |

[a] $NO_{3,\ org}$ for upper bound is denoted as $NO_{3\_org\_ratio\_1}$, and $NO_{3,\ org}$ for lower bound is denoted as $NO_{3\_org\_ratio\_2}$.
[b] $NO_{3,\ org}$ estimated using the PMF method is denoted as $NO_{3\_org\_PMF}$.
To further verify the reliability of the estimated results of organic nitrates, the $NO_{3,\ org}$ concentration time series calculated by
the two methods in each season are shown in Figure 2a. The computed correlation coefficients (R) are good (0.82 for spring,
0.82 for summer and 0.77 for autumn), indicating that similar results were achieved. The inorganic nitrate ($NO_{3\_inorg*}$) obtained
by subtracting $NO_{3\_org\_ratio\_1}$ from total measured nitrates also correlated well with the inorganic nitrate estimated using the
PMF method (R=0.92 for spring, 0.87 for summer and 0.86 for autumn). While they were distinctive from those of inorganic
nitrate (Figure 2b), which indicates that organic nitrates had been well separated from inorganic nitrate in this study, the diurnal
trends of organic nitrates obtained by the two methods were similar in each season, with lower concentrations in the daytime
and higher concentrations at night.

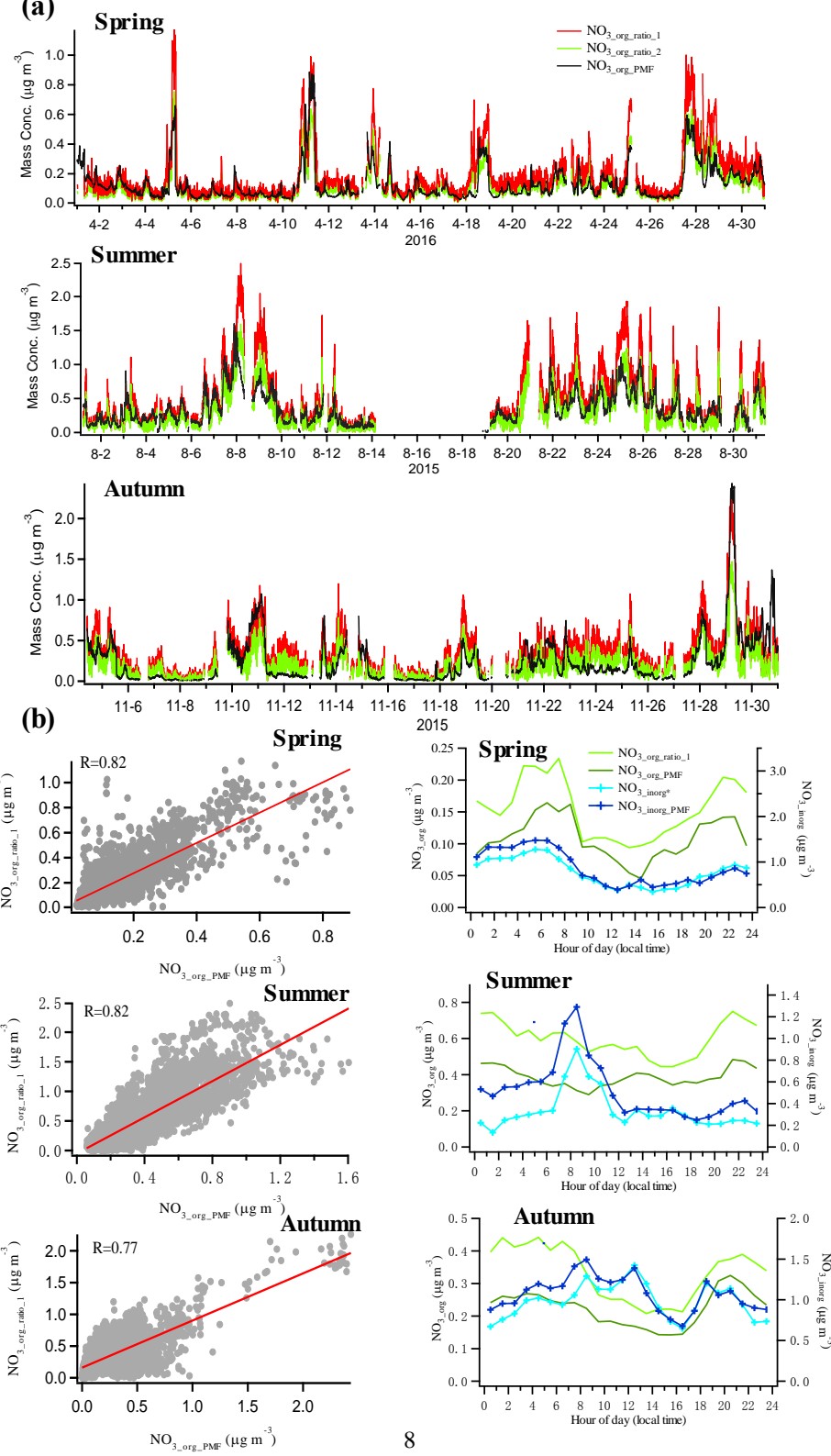

**Figure 2.** (a) Time series of NO$_{3, org}$ concentration estimated by the NO$^+$/NO$_2^+$ ratio method and PMF method for each season; (b) correlations between NO$_{3\_org\_ratio\_1}$ and NO$_{3\_org\_PMF}$ (left panel); diurnal trends of organic nitrates and NO$_{3, org}$ estimated by the different methods (right panel).

## 3.2 Correlation between organic nitrates and OA factors

As indicated by the results in the PMF method, the majority of organic nitrates were associated with LO-OOA in spring, summer and autumn in the urban atmosphere in Shenzhen, implying a dominant secondary origin of organic nitrates. To further confirm this relationship, we made the correlation analysis between organic nitrates estimated by the NO$^+$/NO$_2^+$ ratio method and the three factors resolved by the PMF analysis with only organic mass spectra in the three seasons. Generally, organic nitrates were found better-correlated with LO-OOA (R=0.69-0.77 in Figure 3) than with HOA and MO-OOA (R=0.03-0.69 in Figures S6-S8), which is consistent with the fact that the majority of organic nitrates were associated with LO-OOA in the PMF method. However, the moderate correlation between organic nitrates and HOA implied possibility of direct emissions of organic nitrates. Furthermore, we found a noticeably improved correlation between LO-OOA and organic nitrates at night (19:00-6:00) and a reduced correlation during the daytime (7:00-18:00) in Figure 3, especially in summer, implying that organic nitrates formation might be more closely related to secondary formation at night.

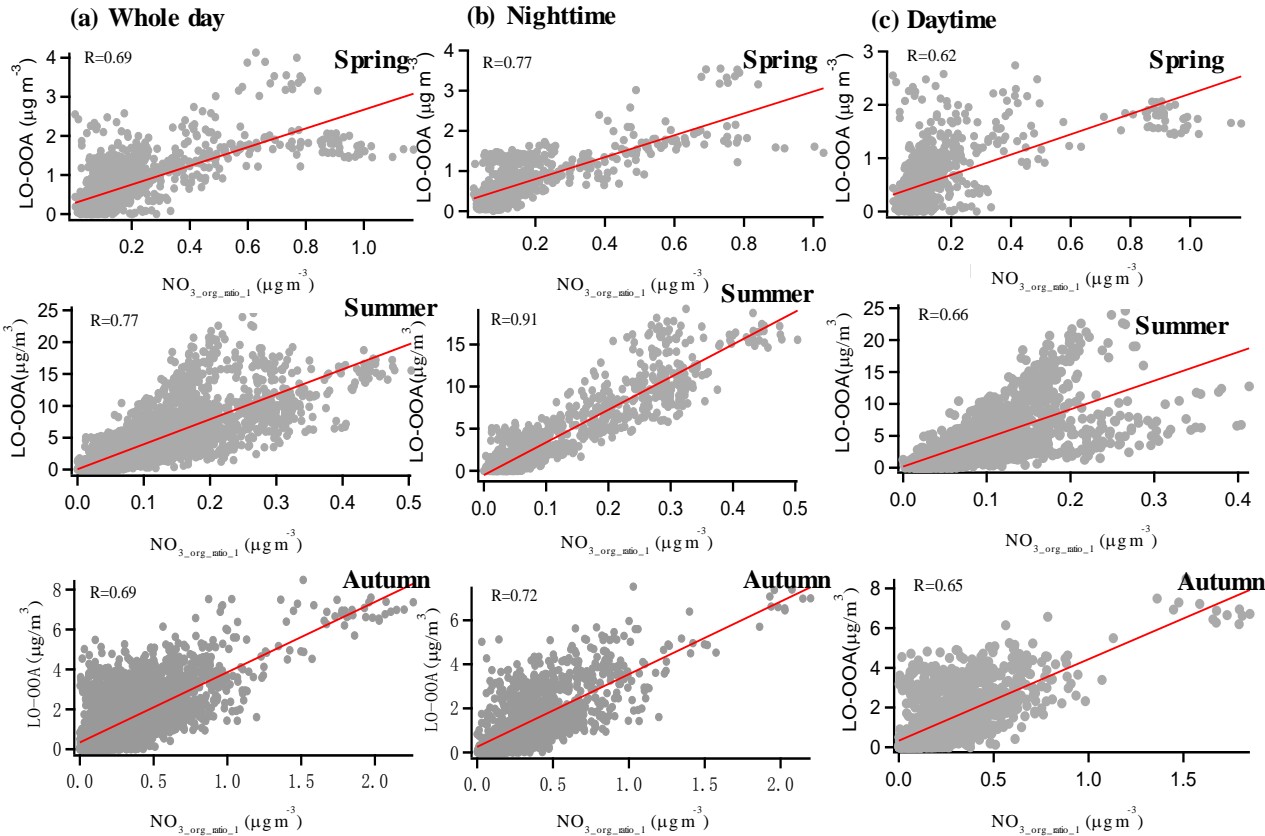

**Figure 3.** Correlation of $NO_{3\_org\_ratio\_1}$ and LO-OOA in each season for the whole day (a), at night (b) and in the daytime (c).
**3.3 Size distribution characteristics of organic nitrates**
In this section, we attempt to use the $NO^+/NO_2^+$ ratio as an indicator to investigate the size distribution characteristics of organic
nitrates. Unfortunately, due to the lack of HR-PToF data, our analyses used the UMR-PToF data of m/z 30 and 46, which
might contain the interferences of $CH_2O_X^+$ (Fry et al., 2018). In our case, the time variations of contributions of $CH_2O^+$ in m/z
and $CH_2O_2^+$ in m/z 46 in the HR data of $PM_1$ for the four seasons are shown in Figure S10. For all the four seasons, the
average contributions of $CH_2O_X^+$ in m/z 30 and 46 in the HR data of $PM_1$ were less than 10%, suggesting that the m/z 30/m/z
ratio could mostly represent the $NO^+/NO_2^+$ ratio. The average size distributions of m/z 30 and m/z 46 for the four seasons
are shown in Figure S11, and Figure 4a shows the average size distributions of different aerosol species and the m/z 30/m/z 46
ratio in the four seasons. It is clearly found that the m/z 30/m/z 46 ratio exhibited a decreasing trend in spring, summer and
autumn, while it kept constant in winter, similar to the value of $R_{NH4NO3}$ (red dotted line in Figure 4a). In addition, in spring,
summer and autumn, the lowest values of the m/z 30/m/z 46 ratio, occurring at ~1 μm, were approximate to the corresponding
seasonal values of $R_{NH4NO3}$. It should be noted that the similar size distribution patterns of the m/z 30/m/z 46 ratio under the
highest interferences (>15%) and lowest interferences (<5%) of $CH_2O_X^+$, indicated by the HR data of $PM_1$, for spring, summer
and autumn (Figure S12) imply that the size distribution patterns of the m/z 30/m/z 46 ratio were not affected significantly by
the interferences of $CH_2O_X^+$. We also used the size distributions of the m/z 30/m/z 46 ratio to separate the size distributions of
inorganic and organic nitrates, as shown in Figure S13, and the results indicate that organic nitrates were relatively more
concentrated at small sizes compared to inorganic nitrates. Furthermore, the diurnal trends of the size distribution of the m/z
30/m/z 46 ratio in spring, summer and autumn in Figure 4b show apparent higher values at small sizes at night, suggesting an
important nighttime local origin of organic nitrates. Combining with the analysis in section 3.2, the local nighttime secondary
formation of organic nitrates in warmer seasons in the urban polluted atmosphere in Shenzhen is highlighted. This is consistent
with the previous findings in the US and Europe that the nighttime $NO_3$+VOCs reactions serve as an important source for
particulate organic nitrates (Rollins et al., 2012; Xu et al., 2015a, 2015b; Fry et al., 2013; Lee et al., 2016). We will then
explore the nighttime $NO_3$+VOCs reactions in Shenzhen in more detail in the following section.

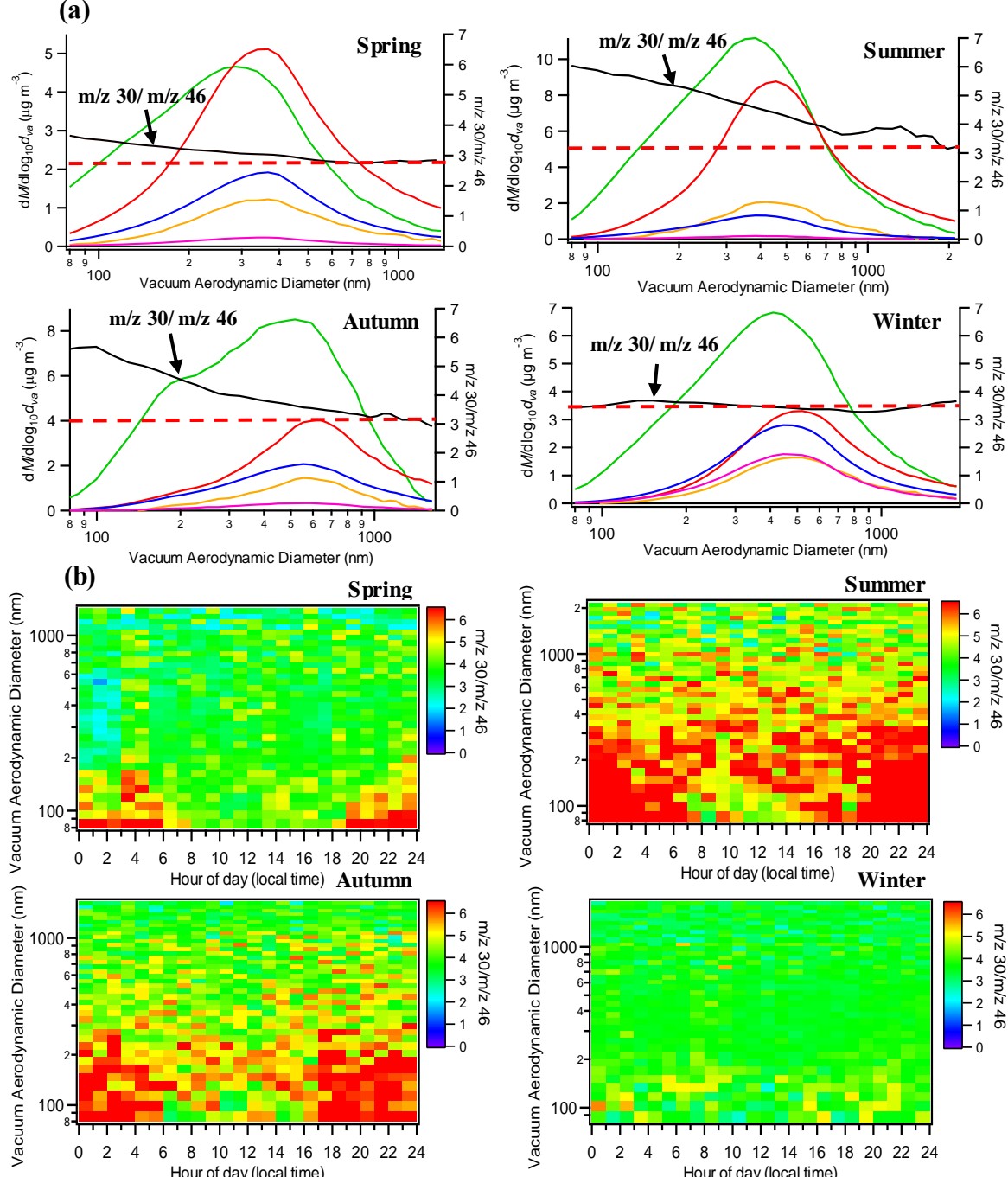


**Figure 4.**(a) Average size distributions of aerosol species and m/z 30/m/z 46 ratio (red dotted line represents $R_{NH4NO3}$); (b)
diurnal trends of size distribution of m/z 30/m/z 46 ratio.

**3.4 Nighttime particulate organic nitrates formation via NO₃+VOCs**

Since on-line VOCs measurement was only performed during the spring campaign (described in section 2.2), the following theoretical analysis of NO₃+VOCs reactions applies only to the spring case. NO₃+VOCs reactions would yield a large mass of gas-and particle-phase organic nitrates (Rollins et al., 2012; Nah et al., 2016; Boyd et al., 2015, 2017; Xu et al., 2015a, 2015b; Lee et al., 2016). We used Eq. (9) to roughly judge the production potential (PP) of organic nitrates from a NO₃+VOC reaction:

$$[\text{Production Potential}]_{NO3+VOCi} = K_i \cdot [VOC_i] \cdot [NO_3] \quad (9)$$

Where $K_i$ represents the reaction rate coefficient for NO₃ radical and a VOC; $[VOC_i]$ is the concentration of the specific VOC; $[NO_3]$ is the concentration of NO₃ radical. It should be noted that no organic nitrates yield parameter was introduced in Eq. (9), because only a few organic nitrate yields for BVOCs were available in the literature (Fry et al., 2014; Ng et al., 2017). However, given the fact that the values of $K_i \cdot [VOC_i] \cdot [NO_3]$ for different VOC species can differ by orders of magnitude, not multiplying the organic nitrate yields (ranging from 0 to 1) would not significantly affect the PP ranking of VOCs. In the spring campaign, the diurnal variations of NO₂, O₃ and estimated NO₃ radical concentrations are shown in Figure S14. It was found that as a comparison to the nighttime NO₃ radical concentration reported in literature in the United States (Rollins et al., 2012; Xu et al., 2015a), high concentration of NO₂ (19.93±2.31 ppb) at night leaded to high yield of NO₃ radical (1.24±0.76 ppt) in Shenzhen, as calculated in Text S1.

The reaction rate coefficients of typical measured nighttime VOC concentrations with NO₃ radical and the production potentials are listed in Table S3 and shown in Figure 5. These VOCs were considered based on their higher ambient concentrations and availability for reaction kinetics with NO₃ radical. According to the distribution of production potential, five biogenic VOCs (BVOCs) (i.e., α-pinene, limonene, camphene, β-pinene and isoprene) and one anthropogenic VOC (styrene) were identified as notable VOC precursors with high production potential, while the sum of production potential from the other VOCs was negligible as shown in Figure 5b.

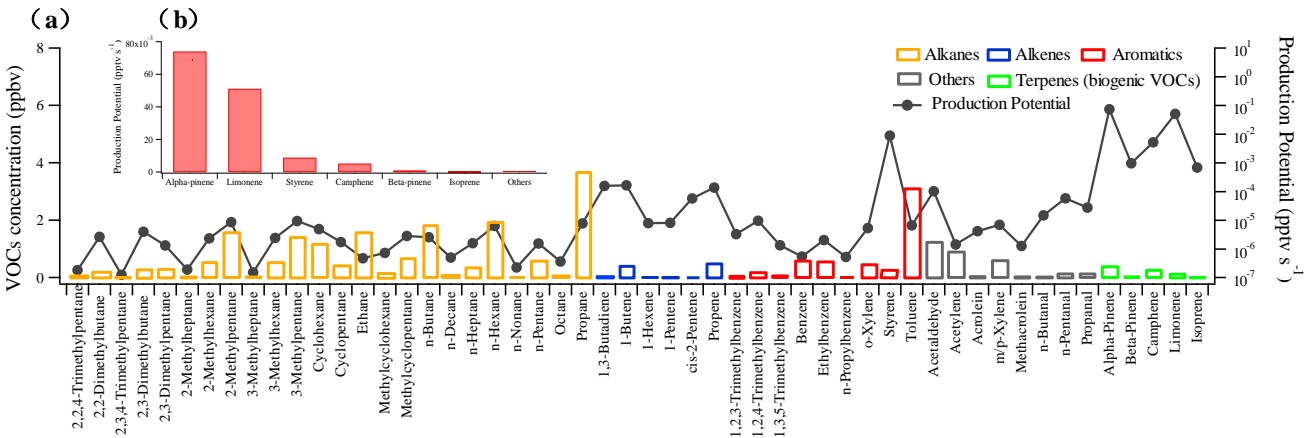

**Figure 5.** (a) Mean concentrations of VOCs and the corresponding calculated production potential of NO₃+VOC at night during the spring campaign; (b) production potential ranking of VOCs at night during the spring campaign.

Based on the production potential evaluation above, we further explore SOA yield of $NO_3$+the six notable VOC precursors
according to the analysis method of particulate organic nitrate formation in Xu et al. (2015a). Briefly, $NO_3$ and ozone are two
main oxidants for SOA formation from VOCs at night. Based on the concentrations of oxidants and the reaction rate constants
for VOCs with $NO_3$ and ozone, the branching ratio of each VOC that reacts with $NO_3$ can be estimated as in Eq. (10). By
combining the estimated branching ratios and SOA yields from chamber studies (Table 3, where the chamber conditions to
obtain the yields covered the range of aerosol mass loading in the spring campaign), potential SOA production from these
VOCs can be calculated as in Eq. (11) (Xu et al., 2015a):
$$branching\ ratio_{species\ i+NO3} = \frac{k_{[species\ i+NO3]} \times [NO3]}{k_{[species\ i+NO3]} \times [NO_3] + k_{[species\ i+O3]} \times [O_3]} \quad (10)$$

$$[SOA]_{species,oxidant} = [species] \times branching\ ratio_{species,oxidant} \times yield_{species,oxidant} \quad (11)$$

The results in Table 3 show that all six notable VOC species were prone to react with $NO_3$ radical instead of $O_3$ at night, and
the estimated potential SOA production from $NO_3$+VOCs reactions using SOA mass yields in the literature was 0-0.33 μg m$^{-}$
$^3$ for α-pinene, 0.09-1.28 μg m$^{-3}$ for limonene, 0.24 μg m$^{-3}$ for styrene, 0.004-0.06 μg m$^{-3}$ for β-pinene and 0.002-0.02 μg m$^{-3}$
for isoprene. The SOA yield from camphene is currently unknown in the literature. It is seen that the average observed
nighttime concentration of particulate organic nitrates during the spring campaign (0.39-0.83μg m$^{-3}$, converting $NO_{3.org\_ratio\_1}$,
$NO_{3.org\_PMF}$ in Figure 6 into organic nitrates assuming the average molecular weight of organic nitrates of 200 to 300 g mol$^{-1}$)
was well within the estimated SOA concentration ranges produced by α-pinene, limonene and styrene in Table 3, indicating
that these three VOCs were the key VOC precursors in urban atmosphere in Shenzhen. Considering both the production
potentials and SOA yields, the contributions of β-pinene and isoprene to nighttime formation of particulate organic nitrates
could be negligible.
**Table 3.** Average concentrations, reaction branching and SOA production of α-pinene, limonene, styrene, camphene, β-
pinene and isoprene with respect to different oxidants at night in the spring campaign.

| Species | Concentration (ppbv) | Rate coefficient [a] | | Branching ratio | | SOA yield from the literature (with $NO_3$) | SOA from VOCs + $NO_3$ (μg m$^{-3}$) |
|---|---|---|---|---|---|---|---|
| | | $NO_3$ | $O_3$ | $NO_3$ | $O_3$ | | |
| α-pinene | 0.39 | 6.64E-12 | 7.2E-17 | 0.962 | 0.038 | 0-0.16[b] | 0-0.33 |
| Limonene | 0.14 | 1.22E-11 | 1.54E-16 | 0.957 | 0.043 | 0.12-1.74[c] | 0.09-1.28 |
| Styrene | 0.19 | 1.50E-12 | 1.70E-17 | 0.941 | 0.059 | 0.23[d] | 0.24 |
| Camphene | 0.28 | 6.20E-13 | 9.0E-19 | 0.992 | 0.008 | / | / |
| β-pinene | 0.01 | 2.51E-12 | 1.50E-17 | 0.968 | 0.032 | 0.07-1.04[e] | 0.004-0.06 |
| Isoprene | 0.032 | 6.96E-13 | 1.27E-17 | 0.908 | 0.091 | 0.02-0.24 [f] | 0.002-0.02 |

[a] Rate coefficients for all species except camphene are from the Master Chemical Mechanism model
(http://mcm.leeds.ac.uk/MCM/; under 25 ℃), rate coefficients for camphene are from Martínez et al. (1999) and Atkinson et
al. (1990).
[b] Hallquist et al. (1999); Spittler et al. (2006); Perraud et al. (2010); Fry et al. (2014).
[c] Fry et al. (2011, 2014); Spittler et al. (2006); Boyd et al. (2017).
[d] Cabrera-Perez et al. (2017).
[e] Griffin et al. (1999); Fry et al. (2009); Fry et al. (2014); Boyd et al. (2015).
[f] Rollins al. (2009); Ng et al. (2008).
The estimation of potential SOA production above suggests significant contributions of α-pinene, limonene, and styrene to
SOA, and the significant contribution of camphene is also possible. Thus, we further explore the diurnal variations of the PPs
of these four VOCs. Figure 6 shows the diurnal trends of BC, LO-OOA, $NO_{3.org\_ratio\_1}$, $NO_{3.org\_PMF}$, and the PPs of the four
VOCs during the spring campaign. There were two apparent nighttime growth periods (i.e., I: 19:00–22:00 and II: 2:00–6:00)
for both $NO_{3.org\_ratio\_1}$ and $NO_{3.org\_PMF}$. During Period I, BC maintained a relatively higher level, suggesting stable anthropogenic
emissions. In contrast, the increases of all the PPs during Period I indicated that these precursors contributed to the organic
nitrate growth. After 22:00, while the PPs still showed a rapid growth, BC and organic nitrates began to decrease, implying
possible existence of other important anthropogenic VOC precursors, which were not identified by the GC-FID/MS analysis
but would dominate the formation of organic nitrates at this stage. During Period II, the anthropogenic emissions remained at
a stable lower level, as indicated by BC, while all the PPs increased with organic nitrates again, indicating that these four
precursors also contributed to, or could dominate, this organic nitrate growth. As shown in Figure S15, organic nitrates
correlated better with the PPs (R=0.63–0.74) than with LO-OOA (R=0.19–0.31) or BC (R=0.02–0.05) during Period II at the
spring campaign, suggesting the significant contributions of the $NO_3$ reactions with these precursors.

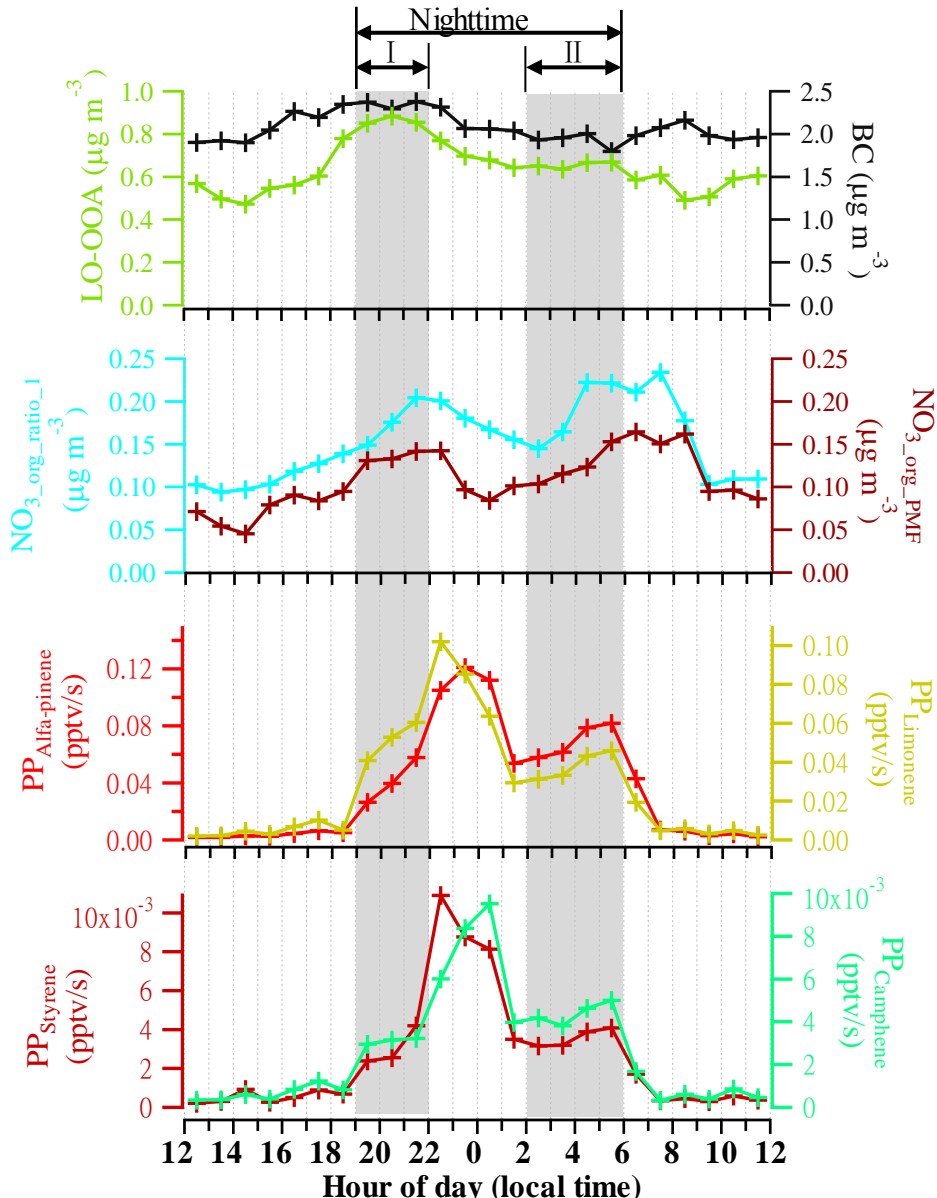


**Figure 6.** Diurnal trends of BC, LO-OOA, $NO_{3.org\_ratio\_1}$, $NO_{3.org\_PMF}$ and production potential (PP) of $\alpha$-pinene, limonene, styrene, and camphene during the spring campaign.

It should be noted that, all previous studies on nighttime organic nitrates formation in the US and Europe focused on mechanisms of $NO_3$ reactions with BVOCs (Hallquist et al., 1999; Spittler et al., 2006; Perraud et al., 2010; Fry et al., 2014; Nah et al., 2016; Boyd et al., 2015, 2017). In this study, however, we found that anthropogenic VOCs could also play significant roles in particulate organic nitrate formation at night. Besides styrene, one of major aromatics (Cabrera-Perez et al., 2016), there were also other important anthropogenic VOC precursors that we did not identify in the spring campaign. In China,

styrene has been actually identified as an important VOC of non-methane hydrocarbons (NMHCs) in urban areas, and has a notable contribution to ozone formation and SOA production (An et al., 2009; Yuan et al., 2013; Zhu et al., 2019). This study highlights the possible key roles of anthropogenic VOC precursors in nighttime particulate organic nitrate formation in urban atmosphere in China, and relevant smog chamber studies for anthropogenic VOCs+$NO_3$ reactions are needed to support parameterization in modeling.

**3.5 Comparison with other similar studies and implications**

Table 4 shows the average ambient temperatures, average concentrations of NO, $NO_2$, monoterpenes, $NO_{3,org}$, the ratio of $NO_{3,org}$ to $NO_{3,total}$ and the ratio of organic nitrates to total organics in several similar field campaigns available in the literature, which implies the key role of $NO_3$+VOCs reactions for nighttime particulate organic nitrate formation. In general, the variation of the particulate organic nitrates concentration is within an order of magnitude (0.06-0.98 $\mu g/m^3$) among the different sites. Higher concentrations of particulate organic nitrates generally are associated with higher NOx concentrations rather than BVOC concentrations. On the other hand, although the BVOCs concentrations in Bakersfield were far less than that in other campaigns, the concentration of particulate organic nitrates there showed an intermediate level among all the campaigns. Therefore, it is suggested that the formation of particulate organic nitrates may be more relevant with NOx than BVOCs, which is consistent with the finding that the organic nitrate production was dominated by NOx in the southeastern US (Edwards et al., 2017). In the spring campaign of this study, we examined the correlation between organic nitrates and $NO_2$ or VOCs (by the sum of α pinene, limonene, camphene and styrene) at night (Figure S16) and found a significant correlation of organic nitrates with $NO_2$ (R=0.40-0.47) rather than with VOCs (R=0.22-0.23), which further suggests that the organic nitrates formation was driven by the NOx-involved $NO_3$ chemistry.

**Table 4.** Average ambient temperatures, average concentrations of monoterpenes, $NO_{3,total}$, $NO_{3,org}$, $NO_{3,org}$/ $NO_{3,total}$ and the ratio of organic nitrates to total organics (ON/Org) for different field campaigns around the world. The ON results at the European and US sites are from Kiendler-Scharr et al. (2016) and Ng et al. (2017).

| Sampling site | Site type | Sampling period | Temperature (˚C) | NO (ppbv) | NO₂ (ppbv) | Monoterpenes (ppbv) | $NO_{3,org}$ (µg m⁻³) | $NO_{3,org}$/ $NO_{3,total}$ | ON/Org | Reference/Note |
|---|---|---|---|---|---|---|---|---|---|---|
| Bakersfield, **US** | rural | May-June, 2010 | 23.0 | | 8.2 | 0.045 (α-pinene) 0.004 (β-pinene) 0.034 (limonene) | 0.16 | 0.28 | 0.23 | Rollins et al. (2012)/ NO3,org measured by TD-LIF |
| Woodland Park, **US** | high attitude | July-August, 2011 | 15.0 | | 1.2 | 0.25 (monoterpene) | 0.06 | 0.86 | 0.09 | Fry et al. (2013)/ Use AMS data to estimate NO3,org |
| Centreville, **US** | rural | June-July, 2013 | 24.7 | 0.1 | 1.1 | 0.350 (α-pinene)* 0.312 (β-pinene)* 0.050 (limonene)* | 0.08 | 1.00 | 0.10 | Xu et al. (2015a) Xu et al. (2015b)/ Use AMS data to estimate NO3,org |
| Barcelona, **Spain** | urban | March, 2009 | 13.3 | 11.0 | 23.6 | 0.423 (monoterpene) | 0.48 | 0.13 | 0.13 | Mohr et al. (2012) Pandolfi et al. (2014) / Use AMS data to estimate NO3,org |
| Shenzhen, **China** | urban | April, 2016 | 24.5 | 8.0 | 19.4 | 0.391 (α-pinene)* 0.013 (β-pinene)* 0.137 (limonene)* | 0.16 | 0.17 | 0.11 | This study/ Use AMS data to estimate NO3,org |

*BVOC concentration at night.

**4. Conclusions**
An Aerodyne HR-ToF-AMS was deployed in urban Shenzhen for about one month per season during 2015–2016 to
characterize particulate organic nitrates with high time resolution. We discovered high mass fractions of organic nitrates in
total organics during warmer seasons, including spring (9-21%), summer (11-25%) and autumn (9-20%), while particulate
organic nitrates were negligible in winter. The correlation analysis between organic nitrates and each OA factor showed high
correlation (R=0.77 in spring, 0.91 in summer and 0.72 in autumn) between organic nitrates and LO-OOA at night. The diurnal
trend analysis of size distribution of m/z 30/m/z 46 ratio further suggested that organic nitrates formation mainly occurred at
night. It also suggested that organic nitrates concentrated at smaller sizes, indicating that they were mostly local products. The
calculated theoretical nighttime production potential of NO$_3$ reacting with VOCs measured in spring showed that six VOC
species (i.e., α-pinene, limonene, styrene, camphene, β-pinene and isoprene) were prominent precursors. The SOA yield
analysis and the nighttime variation of production potential further indicated that α-pinene, limonene, camphene and styrene
could contribute significantly to nighttime formation of particulate organic nitrates in spring in Shenzhen, highlighting the
unique contribution of anthropogenic VOCs as a comparison to that documented in previous studies in the US and Europe.
Finally, the comparison of the results in this study with other similar studies implied that nighttime formation of particulate
organic nitrates is more relevant with NOx levels.
**Author contributions.**
Xiao-Feng Huang designed the research. Kuangyou Yu and Qiao Zhu conducted data analysis and wrote the paper. Ke Du
contributed to modelling and writing.
**Acknowledgments**
This work was supported by National Key R&D Program of China (2018YFC0213901), National Natural Science Foundation
of China (91544215; 41622304) and Science and Technology Plan of Shenzhen Municipality (JCYJ20170412150626172).

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
