# Peer review of "Characterization of nighttime formation of particulate organic"

_Atmospheric Chemistry and Physics, 2018_

## Referee Comment (RC1) · Anonymous Referee #1 · 1 Nov 2018

The authors characterized organic nitrates at an urban site in south China based on the measurements of aerosol mass spectrometer and volatile organic compounds (VOCs). The ratios of $NO^+/NO_2^+$ and positive matrix factorization were used to estimate the concentrations of organic nitrates. The authors found that organic nitrates contribute substantial fractions of total nitrate in spring, summer and autumn, and the reactions between biogenic VOCs and $NO_3$ radical were the major formation pathway. This manuscript is generally well written, and I have some comments below.

Comments:

1. Concerning the PMF results, the authors need to show more diagnostic plots to justify the results. For example, previous study by the same group (He et al., 2011) was able to identify four OA factors in autumn at the same site, while the biomass burning OA was not resolved in this study. The spectrum of LO-OOA in autumn (Figure S1) shows clear *m/z* 60 and 73 signals, suggesting that more factors are needed. In addition, I would suggest the authors checking the changes in $NO^+$, $NO_2^+$ in OA factors across different fpeak values, and give an estimation of uncertainties.

2. High $NO^+$ signal and $NO^+/NO_2^+$ ratio were also observed in HOA spectrum, are they organic nitrates from direct emissions or some other sources. Please calculate the mass fractions of organic nitrates in three OA factors. A major fraction would be expected in LO-OOA, and not surprisingly, organic nitrates were well correlated with LO-OOA.

3. The authors calibrated AMS every two weeks, could the authors show the $NO^+/NO_2^+$ ratio for each calibration to check the stability of the instrument.

4. In the abstract and conclusions, the authors highlight the importance of organic nitrates using its contribution in total nitrates. I would suggest the authors using the mass fractions of organic nitrates in total OA. In addition, could the authors compare the mass concentrations of organic nitrates with previous studies in Pearl River Delta?

5. I suggest the authors adding the time series of non-refractory aerosol species during three seasons in supplementary. This is good for readers to evaluate the sources of organic nitrates. For example, the time series of organic nitrates in spring in Figure 2(a) showed clear plumes, which were very likely from regional transport. Without showing other aerosol species, it is hard to tell.

6. A more detailed description of VOC measurements is needed in experimental methods.

7. Spell out "VOCs" in the abstract, and show slopes in Figure 2(b)

---

## Referee Comment (RC2) · Anonymous Referee #2 · 7 Nov 2018

Review of "Characterization of nighttime formation of particulate organic nitrates based on high-resolution aerosol mass spectrometry in an urban atmosphere in China" by Yu et al.

This study used an HR-ToF-AMS to investigate the particulate organic nitrate (pON) in Shenzhen, China over one-year measurement. The authors applied two methods (i.e., $NO^+/NO_2^+$ ratio and PMF analysis including $NO^+$ and $NO_2^+$ ions) to estimate the concentration of pON nitrate. The fractions of pON in total nitrate in different seasons are reported. Further, it is concluded that biogenic VOCs+nitrate radical is the major source of pON, even though the sampling site is located in polluted urban area. The topic is of interest to the community. Overall, the results are clearly presented and consistent with previous studies. My major concern is that this paper is scientifically correct, but not obviously a significant advance in the field. This study mirrored the analysis procedure from previous publications, but did not emphasize the unique contribution to our knowledge on pON. One interesting point to expand discussions on, as the authors briefly discussed, is that pON concentration in this study is similar to that in the southeastern U.S., a region with lower NOx. A deeper investigation on this comparison may reveal whether the pON formation is VOCs- or NOx-limited across different regions. It is also helpful to contrast to the pON concentration in Europe [*Kiendler-Scharr et al.*, 2016], where the NOx is likely comparable to that in this study. In addition, the diurnal variation of pON (i.e., the increase near 3am) is another interesting point to explore. Overall, I recommend publication after major revision.

Other Comments

1.      The diurnal trends of PMF factors should be included. Please add the diurnal trend of LO-OOA in Figure 5.

2.      Cautions are required when using the method in section 3.3 to estimate the pON formation. To investigate the sources of measured pON, what is really required is the reacted amount of VOCs. The calculated pON, on the other hand, is based on the measured VOCs existing in the atmosphere. Thus, the calculated pON is not directly comparable to measured pON. This analysis can only serve as a ballpark estimation. The conclusion that biogenic VOCs + NO3 is the major source of pON should be toned down.

3.      Figure 3. What is "NO3,org1_ratio"?

4.      Figure 5 and Line 209. What causes the pON increase near 3am?

5.      Line 31. Please add reference to support this statement. Replace "recognize" with "recognized".

6.      Line 99. NO, $NO_2$, $NO_{ON}$, and $NO_{2,ON}$ need superscript "+". This should be revised throughout the manuscript.

7.      Line 131. How is the SA calculated?

8.      Line 149. As a sanity check, are $NO^+$ and $NO_2^+$ exclusively apportioned into NIA in winter?

9.      Line 197. It should be section 2.4, instead of 2.3.

Reference

Kiendler-Scharr, A., et al. (2016), Ubiquity of organic nitrates from nighttime chemistry in the European submicron aerosol, *Geophysical Research Letters*, *43*(14), 7735-7744, doi:10.1002/2016GL069239.

---

## Referee Comment (RC3) · Anonymous Referee #3 · 1 Dec 2018

**Review of Yu et al. "Characterization of nighttime formation of particulate organic nitrates based on high-resolution aerosol mass spectrometry in an urban atmosphere in China"**

https://doi.org/10.5194/acp-2018-1009

Description:

This manuscript describes measurements of OA and organic nitrate aerosol in an urban area in south China using AMS during 4 seasons. The AMS nitrate is separated into organic and inorganic nitrate using 2 methods (NOx+ ratio and PMF) and compared. Organic nitrate aerosol is estimated to be a substantial fraction of total nitrate and of OA during the three warmer seasons. OA is also separated into different source factors using PMF. The organic nitrates are shown to correlate with fresher SOA (LO-OOA from PMF) with slightly better correlations during nighttime. NO3 radical concentrations are calculated based on a steady-stated model of production from O3+NO2 and reaction with two terpenes and conclude that the NO3 radical reaction is the dominant loss of the two terpenes. They model the SOA formation from NO3 + two terpenes and seem to argue that it is consistent with the organic nitrate aerosol measurements and that nighttime NO3 + BVOC is the dominant overall source of particle organic nitrates.

Some of the methods are reasonably well described and follow on methods developed and described in the literature, although additional details are needed for several aspects. However, overall nothing new seems to be offered in terms of method development nor advancement in understanding of atmospheric science (which Ref #2 recognized as well). The conclusions the authors arrive at are not surprising and have been previously published. Thus, it largely comes across as a data report, where measurements were made, previous methods are applied and the scientific analysis/interpretation is conducted in this case with less rigor than prior publications. Thus the value to the literature essentially amounts a report of measurements and simple calculations conducted in a different location. The results section consists of only a few figures and ~2 pages of 1.5-space text, a reflection of the thin-ness of new content. Importantly, some conclusions are overstated with insufficient evidence or even faulty logic presented, so in present form is in fact misleading. Therefore, publication in ACP does not seem appropriate. Details are provided below.

Main Issues:

Text/Clarity:
Text was generally readable but there are lots of grammar errors. A few pervasive errors include the addition or omission of articles (e.g. "the", "a") or plurality when not needed or needed. Reviewing by native English speaker BEFORE submission is recommended. Much of the text is written clearly, while in many cases insufficient details were provided to understand exactly what was done.

PMF details:
I agree with Referee 1 that there is not sufficient detail provided on the PMF (both justification of solutions, as well as summary of results). This should also include the NOx+ ratios for each factor, discussion of the meaning and/or possible biases related to the HOA factor containing nitrates and the inorganic factor having a different NOx+ ratios than calibrations. Correlations of organic nitrates vs all of

the factors should be presented (ideally a version of Fig. 3 for each factor in the supplementary). Simply showing that organic nitrates correlate with LO-OOA does not make a convincing case that they are related since often all concentrations largely increase and decrease at a given sampling site together. Additionally diurnal cycles should be shown for all factors

Organic nitrate quantification:
Evidence is thin to install confidence that the methods for separation of organic nitrates is producing meaningful separation. While the methods have been applied before in other papers, the methods may be prone to substantial error and potentially bias when organic nitrates are a small fraction of total nitrate, as is the case here for all seasons except summertime. Diurnal cycles of the total nitrate and inorganic and organic nitrates calculated by the different methods would be helpful. Showing correlations of both the organic AND inorganic nitrate with the PMF factors may also be informative.

NO3 radical calculation:
The NO3 radical concentration calculation is bewildering. The loss in the steady-state calculation is based on only a-pinene and limonene with no justification for this choice (Section 2.4, Table S1). Then the concentrations are neither reported nor shown, it is unclear if they are calculated for only one fixed value or as a time series. What season was this done for? What season does Table S1 pertain to? This is a critical calculation since the conclusion that NO3 dominates over O3 for BVOC losses and for the SOA modeling. The calculation that NO3 accounts for nearly 100% of the BVOC loss at night is a little surprising and also from which major conclusions of this manuscript flow. For the Xu et al. (2015b) study (referenced in the manuscript), they calculated that only 20%/38% of the reacted a/b-binene was with NO3 at night. Might the NO3 calculated here be biased high since only two VOC losses were considered?

NO3-BVOC SOA modeling and conclusions drawn:
As Referee 2 points out regarding the analysis in Sect. 3.3 on the estimation of pON formation, the sources of measured pON depend on reacted VOCs, not on the amount of VOCs present in the atmosphere. The authors seem to be equating the two. Essentially the authors appear to be calculating the relative amounts of production expected if the sampled airmass was allowed to react to completion with no further emissions. This does not equate to the regional productions since it would systematically underweight more reactive compounds and the two metrics may only be loosely connected.

In general, Section 3.3. is highly undersupported and speculative considering the evidence shown. It is not clear what the modeled SOA (blue trace in Fig. 5) even is. Is that the amount produced per unit time from the model? It cannot be the cumulative production since it increases and decreases (and the model does not have dynamics, dilution, mixing, etc.). In any case, the features of organic nitrates and the ambiguous modeled SOA don't match all that well. Also, it seems likely that the day-to-day variability may be of similar or larger magnitude to the variations in the average diurnal cycle (variability bars such as standard deviations would be helpful here). Potentially the very rough similarities may be an averaging artifact? Thus, the diurnal cycle should be supplemented additional evidence, such as time series of the relevant metrics and correlations plots to make a convincing case that the model may be indeed be representing the key processes and explain the measurements. Also, inclusion of inorganic nitrates together with the organic nitrates (diurnal cycle and other relevant plots), would help make a stronger case that the separation of the organic nitrates is meaningful and robust. Based on this section,

the authors conclude (as stated in the abstract) that BVOC + NO3 at night are the dominant formation pathway of organic nitrates in the polluted atmosphere. This simply has not been demonstrated.

Moreover, it appears that the authors are equating boundary layer concentration with overall regional production importance. As the authors note in Sect. 3.3, the boundary layer is expected to be lower during nighttime. Consequently, the concentrations observed represent a smaller volume of air, so equating lower concentrations during daytime with lower overall (column integrated, regional) importance is faulty logic. BL-effects were not considered here nor production during the daytime modeled, thus no conclusions beyond nighttime boundary layer concentrations and production should be drawn based on this analysis. Yet, this manuscript seems to do just that – an example of the vastly overstated implications claimed.

New results:
Kiendler Scharr et al. (2016), Ng et al. (2017), Xu et al. (2015a, 2015b) and others have reported that at many sites throughout the US and Europe, including polluted urban areas, organic nitrates can be substantial fractions of aerosol nitrates and NO3+BVOC reactions can be an important source. The manuscript fails to make a case for what new information this study from one polluted city provides and how it would add to the body of atmospheric literature.

Regional Implications:
The authors broadly refer to their results as pertaining to "South China" which is a large and diverse areas. Unless there is evidence that this site is generally representative of that geographic area, text and conclusions should be limited to just this one urban area. This is an example of overselling the story without providing the supporting analysis.

**Detailed Comments:**

L32-3: "Play a larger role" for what? Reference?

L40-41: The Rollins et al. (2002) paper demonstrated the application of the aerosol-only organic nitrate measurement. The technique for total nitrates was developed and demonstrated a decade earlier (Day et al., 2002).

L41: "measured" would be better than "obtained"

L66: "to obtain more representative samples" seems vague. Clarify.

L74: "literatures" should not be plural.

L82-4: state if the RIE of ammonium was calibrated or assumed.

L105-6: Site Xu et al. (2015a) since isn't this is exactly what they did?

L107-8: Stating that negative calculated organic nitrates means that concentrations must be low is not very analytically sound. E.g., can't this just mean the method isn't working well or there are large

uncertainties due a variety of possible factors? Please revise to be more precise and inclusive of the possible causes.

L127: Fry et al. 2013 is not the proper reference for heterogeneous N2O5 reaction with aerosol. They just performed the calculation.

L130: Units for velocity are wrong.

L131: Where did the aerosol surface area concentration for the modeling come from? It does not appear that an aerosol sizing instrument was used in the study.

L133-135: Why do the authors include the calculation of the N2O5+H2O gas-phase reaction? The lifetime is 500 years!

L145: "6.82 and 19.38 ppb": Too many significant figures. It would be more useful to report averages and standard deviations.

L159: "may explain" instead? Without any quantitative assessment, it is not justified to say that it does explain it.

L165: "adequate" for what or by what measure? Too vague – needs clarification.

L173-4: Statement points out that the organic nitrate is similar to SE US "even though" BC and NOx are higher in S. China. How is this meaningful? If terpenes are dominantly reacting with NO3 then the production would be largely controlled by the amount of terpenes present - which appear to be quite modest at this location. Statements like this need more context/discussion to be meaningful. Or remove such comparisons if not informative and making a clear point.

L180: "both organic spectra" should instead read "both OOA spectra"?

L180-2:  Unclear. Is this referring to running PMF with and without the NOx ions? Please clarify.

182-3: Please report these correlations (table?) and show the correlations in the SI (i.e. duplicate Fig. 3 for MO-OOA and HOA).

L196-198: Limonene and a-pinene shown to account for 90% of NO3 loss? Assuming the authors are referring to Sect. 2.4 (not 2.3 as written), this is not shown there – or anywhere else. Only 2 compounds were considered according to the text (which references Table S1).

Table 1: "NO3-": Is that inorganic nitrate or total nitrate? If total, then the ionic denotation isn't appropriate.

Figure 5: beta-pinene misspelled.

Table S1: a-pinene SOA yield: Why was this single reference value picked, considering that there is a substantial range reported in the literature? See Table 2 in Ng et al. (2017). A range would seem more appropriate here unless the authors justify why this particular one is more appropriate for this study.

**References Cited:**

Day, D. A., Wooldridge, P. J., Dillon, M. B., Thornton, J. A. and Cohen, R. C.: A thermal dissociation laser-induced fluorescence instrument for in situ detection of $NO_2$, peroxy nitrates, alkyl nitrates, and $HNO_3$, J. Geophys. Res., 107(D5-6), 4046, doi:10.1029/2001JD000779, 2002.

Kiendler-Scharr, A., Mensah, A. A. A., Friese, E., Topping, D., Nemitz, E., Prevot, A. S. H. S. H., Äijälä, M., Allan, J., Canonaco, F., Canagaratna, M., Carbone, S., Crippa, M., Dall Osto, M., Day, D. A. A., De Carlo, P., Di Marco, C. F. F., Elbern, H., Eriksson, A., Freney, E., Hao, L., Herrmann, H., Hildebrandt, L., Hillamo, R., Jimenez, J. L. L., Laaksonen, A., McFiggans, G., Mohr, C., O'Dowd, C., Otjes, R., Ovadnevaite, J., Pandis, S. N. N., Poulain, L., Schlag, P., Sellegri, K., Swietlicki, E., Tiitta, P., Vermeulen, A., Wahner, A., Worsnop, D. and Wu, H.-C. C.: Organic nitrates from night-time chemistry are ubiquitous in the European submicron aerosol, Geophys. Res. Lett., 43(14), doi:10.1002/2016GL069239, 2016.

Lee Ng, N., Brown, S. S. S. S. S., Archibald, A. T. A. T., Atlas, E., Cohen, R. C. R. C., Crowley, J. N. J. N., Day, D. A. D. A. D. A., Donahue, N. M. N. M. N. M., Fry, J. L., Fuchs, H., Griffin, R. J. R. J., Guzman, M. I. M. I., Herrmann, H., Hodzic, A., Iinuma, Y., Kiendler-Scharr, A., Lee, B. H. B. H. B. H., Luecken, D. J. D. J., Mao, J., McLaren, R., Mutzel, A., Osthoff, H. D. H. D., Ouyang, B., Picquet-Varrault, B., Platt, U., Pye, H. O. T. H. O. T., Rudich, Y., Schwantes, R. H. R. H. R. H., Shiraiwa, M., Stutz, J., Thornton, J. A. J. A., Tilgner, A., Williams, B. J. B. J., Zaveri, R. A. R. A., Ng, N. L., Brown, S. S. S. S. S., Archibald, A. T. A. T., Atlas, E., Cohen, R. C. R. C., Crowley, J. N. J. N., Day, D. A. D. A. D. A., Donahue, N. M. N. M. N. M., Fry, J. L., Fuchs, H., Griffin, R. J. R. J., Guzman, M. I. M. I., Herrmann, H., Hodzic, A., Iinuma, Y., Jimenez, J. L., Kiendler-Scharr, A., Lee, B. H. B. H. B. H., Luecken, D. J. D. J., Mao, J., McLaren, R., Mutzel, A., Osthoff, H. D. H. D., Ouyang, B., Picquet-Varrault, B., Platt, U., Pye, H. O. T. H. O. T., Rudich, Y., Schwantes, R. H. R. H. R. H., Shiraiwa, M., Stutz, J., Thornton, J. A. J. A., Tilgner, A., Williams, B. J. B. J., Zaveri, R. A. R. A., Hermann, H., Hodzic, A., Iinuma, Y., Jimenez, J. L., Kiendler-Scharr, A., Lee, B. H. B. H. B. H., Luecken, D. J. D. J., Mao, J., McLaren, R., Mutzel, A., Osthoff, H. D. H. D., Ouyang, B., Picquet-Varrault, B., Platt, U., Pye, H. O. T. H. O. T., Rudich, Y., Schwantes, R. H. R. H. R. H., Shiraiwa, M., Stutz, J., Thornton, J. A. J. A., Tilgner, A., Williams, B. J. B. J., Zaveri, R. A. R. A., Herrmann, H., Hodzic, A., Iinuma, Y., Jimenez, J. L., Kiendler-Scharr, A., Lee, B. H. B. H. B. H., Luecken, D. J. D. J., et al.: Nitrate radicals and biogenic volatile organic compounds: oxidation, mechanisms, and organic aerosol, Atmos. Chem. Phys., 17(3), 2103–2162, doi:10.5194/acp-17-2103-2017, 2017.

Xu, L., Suresh, S., Guo, H., Weber, R. J. and Ng, N. L.: Aerosol characterization over the southeastern United States using high-resolution aerosol mass spectrometry: spatial and seasonal variation of aerosol composition and sources with a focus on organic nitrates, Atmos. Chem. Phys., 15(13), 7307–7336, doi:10.5194/acp-15-7307-2015, 2015a.

Xu, L., Guo, H., Boyd, C. M., Klein, M., Bougiatioti, A., Cerully, K. M., Hite, J. R., Isaacman-VanWertz, G., Kreisberg, N. M., Knote, C., Olson, K., Koss, A., Goldstein, A. H., Hering, S. V, de Gouw, J., Baumann, K., Lee, S.-H., Nenes, A., Weber, R. J. and Ng, N. L.: Effects of anthropogenic emissions on aerosol formation from isoprene and monoterpenes in the southeastern United States, Proc. Natl. Acad. Sci., 112(1), 37–42, doi:10.1073/pnas.1417609112, 2015b.

---

## Author Comment (AC1) · 12 Jan 2019

**Report #1**

The authors characterized organic nitrates at an urban site in south China based on the measurements of aerosol mass spectrometer and volatile organic compounds (VOCs). The ratios of NO+/NO2+ and positive matrix factorization were used to estimate the concentrations of organic nitrates. The authors found that organic nitrates contribute substantial fractions of total nitrate in spring, summer and autumn, and the reactions between biogenic VOCs and NO3 radical were the major formation pathway. This manuscript is generally well written, and I have some comments below.

1. Concerning the PMF results, the authors need to show more diagnostic plots to justify the results. For example, previous study by the same group (He et al., 2011) was able to identify four OA factors in autumn at the same site, while the biomass burning OA was not resolved in this study. The spectrum of LO-OOA in autumn (Figure S1) shows clear m/z 60 and 73 signals, suggesting that more factors are needed. In addition, I would suggest the authors checking the changes in NO+, NO2+ in OA factors across different fpeak values, and give an estimation of uncertainties.

**REPLY:**

According to the diagnostic plots of the PMF analysis shown in Figure S2 to S4 in the supplementary, the same organic factors were obtained in PMF based on only OA spectra and OA spectra combining with NOX+ ions. Although the 3-factor solution for OA seemed to have a "mixed factor" problem (Zhu et al., 2018), such as HOA mixed with COA (clear  $C_3H_3O^+$  in m/z 55 for spring, summer and autumn) (Mohr et al., 2012) and BBOA mixed with LO-OOA (clear m/z 60 and 73 signals in LO-OOA for autumn) (Cubison et al., 2011), running PMF with more factors would produce unexplained factors but little influence the apportion between organic nitrate and inorganic nitrate (Table S1). In addition, the uncertainties of NO+ and NO2+ in OA factors across different fpeak values (from –1.0 to 1.0) were very small (Table S2). Thus, the 3-factor solution was finally used for quantifying organic nitrates in spring, summer and autumn. We have added the related description in section 2.3.

Table S1. The correlation coefficients (R) of NO3, org and NO3, inorg in 3- to 5-factors solutions estimated

|        |       | 3-factor solution   |                       | 4-factor solution   |                       | 5-factor solution   |                       |
|--------|-------|---------------------|-----------------------|---------------------|-----------------------|---------------------|-----------------------|
|        |       | NO 3,org | NO 3,inorg | NO 3,org | NO 3,inorg | NO 3,org | NO 3,inorg |
|        |       | (NOx vs.PMF)        | (NOx vs.PMF)          | (NOx vs.PMF)        | (NOx vs.PMF)          | (NOx vs.PMF)        | (NOx vs.PMF)          |
| Spring | R     | 0.82                | 0.92                  | 0.81                | 0.90                  | 0.80                | 0.91                  |
|        | Slope | 1.21                | 0.76                  | 1.15                | 0.78                  | 1.20                | 0.82                  |
| Summer | R     | 0.82                | 0.87                  | 0.82                | 0.88                  | 0.81                | 0.90                  |
|        | Slope | 1.53                | 0.70                  | 1.50                | 0.65                  | 1.45                | 0.64                  |
| Autumn | R     | 0.77                | 0.86                  | 0.75                | 0.85                  | 0.76                | 0.83                  |
|        | Slope | 0.81                | 0.85                  | 0.76                | 0.82                  | 0.75                | 0.78                  |

by PMF method with these estimated by NOx method, respectively.

|        |                              |             | НОА                   | LO-OOA               | MO-OOA               |
|--------|------------------------------|-------------|-----------------------|----------------------|----------------------|
| Spring | $NO^+$                       | fpeak=0     | 1.3*10 -2  | 1.4*10-2             | 9.8*10 -3 |
|        |                              | uncertainty | 3.4%                  | 1.4%                 | 6.8%                 |
|        | NO 2 + | fpeak=0     | 1.2*10 -2  | 1.5*10-4             | 3.0*10 -8 |
|        |                              | uncertainty | 8.5%                  | 3.8%                 | 4.3%                 |
| Summer | NO +              | fpeak=0     | 1.5*10-2              | 1.0*10-2             | 1.2*10-2             |
|        |                              | uncertainty | 1.5%                  | 5.1%                 | 3.0%                 |
|        | $NO_2^+$                     | fpeak=0     | 1.47*10 -6 | 6.7*10-4             | 1.8*10-3             |
|        |                              | uncertainty | 4.8%                  | 6.9%                 | 4.0%                 |
| Autumn | $NO^+$                       | fpeak=0     | 1.1*10 -2  | 3.1*10-2             | 1.0*10-2             |
|        |                              | uncertainty | 4.5%                  | 0.5%                 | 1.2%                 |
|        | NO 2 + | fpeak=0     | 7.0*10 -8  | 9.8*10 -8 | 2.8*10 -7 |
|        |                              | uncertainty | 0.6%                  | 1.9%                 | 1.2%                 |

**Table S2.** The mass fraction of NO+ and NO2+ in OA factors when fpeak is 0 and the uncertainties of NO+ and NO2+ in OA factors across different fpeak values (from -1.0 to 1.0)

2. High NO+ signal and NO+/NO2+ ratio were also observed in HOA spectrum, are they organic nitrates from direct emissions or some other sources. Please calculate the mass fractions of organic nitrates in three OA factors. A major fraction would be expected in LO-OOA, and not surprisingly, organic nitrates were well correlated with LO-OOA.

**REPLY:**

The mass fraction of organic nitrates in HOA, LO-OOA and MO-OOA was 31%, 49% and 20%, respectively, in spring; 28%, 52% and 20%, respectively, in summer; 30%, 46% and 24% in HOA, LO-OOA and MO-OOA, respectively, in autumn. We have added this statement in section 3.1.

3. The authors calibrated AMS every two weeks, could the authors show the NO+/NO2+ ratio for each calibration to check the stability of the instrument.

**REPLY:**

In this study,  $R_{NH4NO3}$  was determined by IE calibration using pure NH4NO3 on every two weeks for each campaigns and the results show stable values: In spring, the average  $R_{NH4NO3}$  was 2.66 for the first IE calibration and 2.94 for the second IE calibration; in summer, the average  $R_{NH4NO3}$  was 3.05 and 3.34 for the first and second IE calibration, respectively; in autumn, the average  $R_{NH4NO3}$  was 3.33 and 3.31 for the first and second IE calibration, respectively; in winter, the average  $R_{NH4NO3}$  was 3.45 and 3.51 for the first and second IE calibration, respectively. We have added this in section 2.3.

4. In the abstract and conclusions, the authors highlight the importance of organic nitrates using its

contribution in total nitrates. I would suggest the authors using the mass fractions of organic nitrates in total OA. In addition, could the authors compare the mass concentrations of organic nitrates with previous studies in Pearl River Delta?

**REPLY:**

We have amended the description in the abstract and conclusions according to this comment. According to the best of our knowledge, no literature has reported the mass concentrations of organic nitrates in Pearl River Delta region, but we have added section 3.5 to compare the organic nitrates in this study to other similar studies across the world.

5. I suggest the authors adding the time series of non-refractory aerosol species during three seasons in supplementary. This is good for readers to evaluate the sources of organic nitrates. For example, the time series of organic nitrates in spring in Figure 2(a) showed clear plumes, which were very likely from regional transport. Without showing other aerosol species, it is hard to tell.

**REPLY:**

We have added the time series of non-refractory aerosol species in Figure S1 in the supplementary.

6. A more detailed description of VOC measurements is needed in experimental methods.

**REPLY:**

We have added the related description in section 2.2.2. "In the spring campaign, ambient VOC concentrations were also measured using an on-line VOC monitoring system (TH-300B, Tianhong Corp.), including an ultralow-temperature preconcentration cold trap and an automated in-situ gas chromatograph (Agilent 7820A) equipped with a mass spectrometer (Agilent 5977E). The system had both a flame ionization detector (FID) gas channel for C2–C5 hydrocarbons and a mass spectrometer (MS) gas channel for C5–C12 hydrocarbons, halohydrocarbons and oxygenated VOCs. A complete working cycle of the system was one hour and included five steps: sample collection, freeze-trapping, thermal desorption, GC-FID/MS analysis, heating and anti-blowing purification. The sample collection time was 5 min, the sampling flow was 60 ml min-1, and the anti-blowing flow was 200 ml min-1. The calibration of over 100 VOCs was performed using mixed standard gas before and after the campaign. Detection limits for most compounds were near 5pptv. More description of this instrument can be found in Wang et al. (2014)."

7. Spell out "VOCs" in the abstract, and show slopes in Figure 2(b).

**REPLY:**

We have amended it and the slopes were given in Table S1.

**References**

Wang, M.; Zeng, L.; Lu, S.; Shao, M.; Liu, X.; Yu, X.; Chen, W.; Yuan, B.; Zhang, Q.; Hu, M.; Zhang, Z.: Development and validation of a cryogen-free automatic gas chromatograph system (GC-MS/FID) for online measurements of volatile organic compounds, 6, (23), 9424-9434. Analytical Methods, https://doi.10.1039/C4AY01855A, 2014.

---

## Author Comment (AC2) · 12 Jan 2019

Report #2

This study used an HR-ToF-AMS to investigate the particulate organic nitrate (pON) in Shenzhen, China over one-year measurement. The authors applied two methods (i.e., NO+/NO2+ ratio and PMF analysis including NO+ and NO2+ ions) to estimate the concentration of pON nitrate. The fractions of pON in total nitrate in different seasons are reported. Further, it is concluded that biogenic VOCs+nitrate radical is the major source of pON, even though the sampling site is located in polluted urban area. The topic is of interest to the community. Overall, the results are clearly presented and consistent with previous studies.

1. My major concern is that this paper is scientifically correct, but not obviously a significant advance in the field. This study mirrored the analysis procedure from previous publications, but did not emphasize the unique contribution to our knowledge on pON. One interesting point to expand discussions on, as the authors briefly discussed, is that pON concentration in this study is similar to that in the southeastern U.S., a region with lower NOx. A deeper investigation on this comparison may reveal whether the pON formation is VOCs- or NOx-limited across different regions. It is also helpful to contrast to the pON concentration in Europe [*Kiendler-Scharr et al.*, 2016], where the NOx is likely comparable to that in this study. In addition, the diurnal variation of pON (i.e., the increase near 3am) is another interesting point to explore.

**REPLY:**

We have added some description in *Introduction* to address the contribution of this paper on organic nitrates studies in detail. "Ng et al. (2017) reviewed the nitrate radical chemistry and the abundance of particulate organic nitrates in the United States and Europe, and further concluded that particulate organic nitrates are formed substantially via $NO_3$+BVOC chemistry, which plays an important role in SOA formation. Unfortunately, relevant Chinese datasets are scarce yet and not included in this review. This was because (1) the contributions of organic nitrates in SOA and total nitrates in Chinese atmosphere remain poorly understood; (2) the anthropogenic and biogenic precursor emissions in China are largely different from those in the United States and Europe, and thus cannot be easily estimated. To our best knowledge, few studies have investigated the concentrations and formation pathways of particulate organic nitrates in China. Xu et al. (2017) estimated the mass concentration of organic nitrogen in Beijing using AMS, but in this study they ignored the contribution of $NO_X^+$ family, which are the major fragments of organic nitrates." In addition, we added section 3.3 to discuss the size distribution characteristics of organic nitrates. We used the $NO^+/NO_2^+$ ratio as an indicator to investigate the size distribution characteristics of organic nitrates. It is clearly found that the $NO^+/NO_2^+$ ratio generally increases towards smaller size in spring, summer, and autumn, while the $NO^+/NO_2^+$ ratio keep similar to the value of $R_{NH4NO3}$ throughout the full size range in winter. It should also be noted that in spring, summer, and autumn, the lowest values of $NO^+/NO_2^+$ ratio at > 1μm are also approximate to the seasonal values of $R_{NH4NO3}$. These characteristics clearly indicate that organic nitrates occurred mostly in fresh particles with smaller sizes, and thus should be mainly of local origin. The diurnal trends of size distributions of $NO^+/NO_2^+$ ratio show higher values in small size range at night comparing that in the day in spring, summer and autumn, suggesting a dominant nighttime origin of organic nitrates. Furthermore, according to the calculated production potential of organic nitrates from a $NO_3$+VOC reaction and SOA yields in section 3.4, we find that besides the BVOCs species, anthropogenic styrene plays an important role in nighttime particulate organic nitrate formation in urban atmosphere in China. Finally, we compared this study with other particulate organic nitrate studies in section 3.5 and the results show that the formation of particulate organic nitrate is more likely

NOx-control than BVOCs-control and high NOx emissions could promote biogenic SOA formation at night. The detailed reply regarding the diurnal variation of pON can be found in Question 5 below.

2. The diurnal trends of PMF factors should be included. Please add the diurnal trend of LO-OOA in Figure 5.

**REPLY:**

We have added the diurnal trend of LO-OOA in Figure 6.

3. Cautions are required when using the method in section 3.3 to estimate the pON formation. To investigate the sources of measured pON, what is really required is the reacted amount of VOCs. The calculated pON, on the other hand, is based on the measured VOCs existing in the atmosphere. Thus, the calculated pON is not directly comparable to measured pON. This analysis can only serve as a ballpark estimation. The conclusion that biogenic VOCs + NO3 is the major source of pON should be toned down.

**REPLY:**

We have made a lot of modification in the manuscript and tone down the conclusion that biogenic VOCs + NO3 is the major source of pON. According to the section 3.2 and 3.3, we can conclude the nighttime NO$_3$+VOCs reactions serve as an important source for organic nitrates, and in section 3.4, further investigation on potential pathway for nighttime particulate organic nitrates formation was performed. We used the NO$_3$ loss rate at night, which can be calculated as $K_i \cdot [VOC_i]$ in Eq. (9), to roughly judge the production potential of organic nitrates from a NO$_3$+VOC reaction:

$$[\text{Production Potential}]_{NO3+VOCi} = K_i \cdot [VOC_i] \cdot [NO_3] \quad (9)$$

Where $K_i$ represents the reaction rate coefficient for NO$_3$ radical and a VOC, $[VOC_i]$ is the concentration of the specific VOC and $[NO_3]$ is the concentration of NO$_3$ radical. According to the distribution of production potential, five biogenic VOCs (BVOCs) (i.e., α-pinene, limonene, camphene, β-pinene and isoprene) and one anthropogenic VOC (styrene) were identified as notable VOC precursors with high production potential, while the sum of production potential from the other VOCs was negligible. In addition, the estimated SOA production from NO$_3$+VOCs reactions using SOA mass yields shows that α-pinene, limonene and styrene were the key VOC precursors for nighttime organic nitrates formation in urban atmosphere in Shenzhen. This result highlights the key role of this anthropogenic VOC precursor in nighttime particulate organic nitrate formation in urban atmosphere in China, and relevant smog chamber studies for anthropogenic VOCs+NO3 reactions are needed to support parameterization in modeling.

4. Figure 3. What is "NO3,org1_ratio"?

**REPLY:**

The NO3 estimation based on NO$^+$/NO$_2^+$ ratio method using the upper bounds of R$_{ON}$/R$_{NH4NO3}$ is denoted as NO$_{3.org1\_ratio}$, Please see the footnote in Table 2.). In order to see it clearly, we changed it to "NO$_{3\_org\_ratio\_1}$" in the text.

5. Figure 5 and Line 209. What causes the pON increase near 3am?

**REPLY:**

We have added the detailed discussion in section 3.4 to explain the pON increase near 3am. "Figure 6 shows the average nighttime variations of BC, LO-OOA, NO$_{3.org\_ratio\_1}$, NO$_{3.org\_PMF}$ and production

potential of the six notable VOCs identified during the spring campaign. The concentrations of BC and LO-OOA generally decreased slowly after sunset till sunshine due to the combined effect of both the planetary boundary layer variation and traffic emissions, while particulate organic nitrates showed a different trend with two clear growth processes (19:00-22:00 and 3:00-6:00) at night, suggesting their unique sources. In contrast, the production potentials of the six notable VOCs with $NO_3$ had two roughly similar increases at the same periods as those of particulate organic nitrates, which supported the key role of $NO_3$+VOCs reactions for nighttime organic nitrate formation."

6. Line 31. Please add reference to support this statement. Replace "recognize" with "recognized".

**REPLY:**

We have added the reference and replaced "recognized" with "recognize".

7. Line 99. NO, NO2, NOON, and NO2, ON need superscript "+". This should be revised throughout the manuscript.

**REPLY:**

We have corrected it.

8. Line 131. How is the SA calculated?

**REPLY:**

SA is calculated from the size-resolved particle number concentrations assuming spherical particles measured by a scanning mobility particle sizer (SMPS) (TSI Inc., USA, 3775 CPC and TSI Inc. 3080 DMA). And 220 $\mu m^2 cm^{-3}$ is actually under dry conditions, the ambient (wet) aerosol SA is 475 $\mu m^2 cm^{-3}$ by using the hygroscopic growth factor in Liu et al. (2010). We have added and corrected the related description in Text S1 in the supplementary. We have added this description in Text S1 in the supplementary.

9. Line 149. As a sanity check, are NO+ and NO2+ exclusively apportioned into NIA in winter?

**REPLY:**

The $NO_X^+$ method is not suitable to estimate the mass concentration of organic nitrates due to more than 50% ambient $NO^+/NO_2^+$ values smaller than $R_{NH4NO3}$. Thus, we use PMF method to estimate the organic nitrates and the result are shown in the figure (added to FigureS6) below. A majority $NO^+$ and $NO_2^+$ ions are apportioned into NIA and the contribution of organic nitrates in total nitrates is about 4%. We have added some description in Section 3.1. "It should be noted that the small difference between the average $R_{obs}$ and $R_{NH4NO3}$ in winter leads to a large portion of negative data using the $NO^+/NO_2^+$ ratio method (Table 1), and the result from the PMF method shows the contribution of organic nitrates in total nitrates is only 4.2% in winter (Figure S6), suggesting a negligible contribution of organic nitrates. Thus, we will only discuss organic nitrate estimation results in spring, summer and autumn."

10. Line 197. It should be section 2.4, instead of 2.3.

**REPLY:**

We have corrected it.

**References**

Liu, X.G., Zhang, Y.H., Wen, M.T., Wang, J.L., Jung, J.S., Chang, S.-Y., Hu, M., Zeng, L.M. and Kim, Y.J.: A closure study of aerosol hygroscopic growth factor during the 2006 Pearl River Delta Campaign. Adv. Atmos. Sci. 27, 947–956, 2010.

Ng, N. L., Brown, S. S., Archibald, A. T., Atlas, E., Cohen, R. C., Crowley, J. N., Day, D. A., Donahue, N. M., Fry, J. L., Fuchs, H., Griffin, R. J., Guzman, M. I., Herrmann, H., Hodzic, A., Iinuma, Y., Jimenez, J. L., Kiendler-Scharr, A., Lee, B. H., Luecken, D. J., Mao, J., McLaren, R., Mutzel, A., Osthoff, H. D., Ouyang, B., Picquet-Varrault, B., Platt, U., Pye, H. O. T., Rudich, Y., Schwantes, R. H., Shiraiwa, M., Stutz, J., Thornton, J. A., Tilgner, A., Williams, B. J., and Zaveri, R. A.: Nitrate radicals and biogenic volatile organic compounds: oxidation, mechanisms, and organic aerosol, Atmos. Chem. Phys., 17, 2103-2162, https://doi.org/10.5194/acp-17-2103-2017, 2017.

Xu, W., Sun, Y., Wang, Q., Du, W., Zhao, J., Ge, X., Han, T., Zhang, Y., Zhou, W., Li, J., Fu, P., Wang, Z., Worsnop, D.R.: Seasonal Characterization of Organic Nitrogen in Atmospheric Aerosols Using High Resolution Aerosol Mass Spectrometry in Beijing, China. ACS Earth Sp. Chem. 1, 673–682. https://doi.org/10.1021/acsearthspacechem.7b00106, 2017.

---

## Author Comment (AC3) · 12 Jan 2019

Report #3

1. Some of the methods are reasonably well described and follow on methods developed and described in the literature, although additional details are needed for several aspects. However, overall nothing new seems to be offered in terms of method development nor advancement in understanding of atmospheric science (which Ref #2 recognized as well). The conclusions the authors arrive at are not surprising and have been previously published. Thus, it largely comes across as a data report, where measurements were made, previous methods are applied and the scientific analysis/interpretation is conducted in this case with less rigor than prior publications. Thus the value to the literature essentially amounts a report of measurements and simple calculations conducted in a different location. The results section consists of only a few figures and ~2 pages of 1.5-space text, a reflection of the thin-ness of new content. Importantly, some conclusions are overstated with insufficient evidence or even faulty logic presented, so in present form is in fact misleading.

**REPLY:**

We have done lots of modification and corrected the mistakes based on each comment to offer more solid and convincing conclusions. Please see the details as below.

2. Text was generally readable but there are lots of grammar errors. A few pervasive errors include the addition or omission of articles (e.g. "the", "a") or plurality when not needed or needed. Reviewing by native English speaker BEFORE submission is recommended. Much of the text is written clearly, while in many cases insufficient details were provided to understand exactly what was done.

**REPLY:**

We have corrected them in the text.

3. I agree with Referee 1 that there is not sufficient detail provided on the PMF (both justification of solutions, as well as summary of results). This should also include the NOx+ ratios for each factor, discussion of the meaning and/or possible biases related to the HOA factor containing nitrates and the inorganic factor having a different NOx+ ratios than calibrations. Correlations of organic nitrates vs all of the factors should be presented (ideally a version of Fig. 3 for each factor in the supplementary). Simply showing that organic nitrates correlate with LO-OOA does not make a convincing case that they are related since often all concentrations largely increase and decrease at a given sampling site together. Additionally diurnal cycles should be shown for all factors

**REPLY:**

The key diagnostic plots of chosen factors and the mass spectrum profiles of 3 to 5 factors for spring, summer and autumn are shown in Figure S2 to S4 (in the supplementary). The $NOx^+$ ratios for NIAs are given in section 3.1 in the manuscript (2.93 for spring, 3.53 for summer and 3.54 for autumn), and $NOx^+$ ratios for OA factors are shown in Table S2 in the supplement. And the uncertainties of $NO^+$ and $NO_2^+$ in OA factors across different fpeak values are also shown in Table S2. Correlations of organic nitrates vs. all of factors are presented in Figure S7 to S9 in the supplementary. The diurnal cycles for OA factors in each season are shown in Figure S5 and it shows that all OA factors have distinctive variation trends in different seasons.

**Table S2.** The values of $NO^+$ and $NO_2^+$ in OA factors when fpeak is 0 and the uncertainties of $NO^+$ and $NO_2^+$ in OA factors across different fpeak values (from −1.0 to 1.0)

| | | | HOA | LO-OOA | MO-OOA |
|---|---|---|---|---|---|
| **Spring** | $NO^+$ | fpeak=0 | $1.3*10^{-2}$ | $1.4*10^{-2}$ | $9.8*10^{-3}$ |

| | | | | | |
|---|---|---|---|---|---|
| | | uncertainty | 3.4% | 1.4% | 6.8% |
| | $NO_2^+$ | fpeak=0 | $1.2*10^{-2}$ | $1.5*10^{-4}$ | $3.0*10^{-8}$ |
| | | uncertainty | 8.5% | 3.8% | 4.3% |
| **Summer** | $NO^+$ | fpeak=0 | $1.5*10^{-2}$ | $1.0*10^{-2}$ | $1.2*10^{-2}$ |
| | | uncertainty | 1.5% | 5.1% | 3.0% |
| | $NO_2^+$ | fpeak=0 | $1.47*10^{-6}$ | $6.7*10^{-4}$ | $1.8*10^{-3}$ |
| | | uncertainty | 4.8% | 6.9% | 4.0% |
| **Autumn** | $NO^+$ | fpeak=0 | $1.1*10^{-2}$ | $3.1*10^{-2}$ | $1.0*10^{-2}$ |
| | | uncertainty | 4.5% | 0.5% | 1.2% |
| | $NO_2^+$ | fpeak=0 | $7.0*10^{-8}$ | $9.8*10^{-8}$ | $2.8*10^{-7}$ |
| | | uncertainty | 0.6% | 1.9% | 1.2% |

4. Evidence is thin to install confidence that the methods for separation of organic nitrates is producing meaningful separation. While the methods have been applied before in other papers, the methods may be prone to substantial error and potentially bias when organic nitrates are a small fraction of total nitrate, as is the case here for all seasons except summertime. Diurnal cycles of the total nitrate and inorganic and organic nitrates calculated by the different methods would be helpful. Showing correlations of both the organic AND inorganic nitrate with the PMF factors may also be informative.

**REPLY:**

We have added more discussion to support that the separation between organic and inorganic nitrates is meaningful in section 3.1. Diurnal cycles of total nitrate and inorganic and organic nitrates calculated by the different methods are given in Figure 2b in the manuscript. The inorganic nitrate ($NO_{3\_inorg*}$) obtained by subtracting $NO_{3\_org\_ratio\_1}$ from total measured nitrates also correlated well with the inorganic nitrate estimated using the PMF method (R=0.92 for spring, 0.87 for summer and 0.86 for autumn). Furthermore, the diurnal trends of organic nitrates obtained by the two methods were also similar in each season, generally with lower concentrations in the daytime and higher concentrations at night, while they were distinctive from those of inorganic nitrate (Figure 2b), supporting that organic nitrates had been well separated from inorganic nitrate in this study.

5. The NO3 radical concentration calculation is bewildering. The loss in the steady-state calculation is based on only a-pinene and limonene with no justification for this choice (Section 2.4, Table S1). Then the concentrations are neither reported nor shown, it is unclear if they are calculated for only one fixed value or as a time series. What season was this done for? What season does Table S1 pertain to? This is a critical calculation since the conclusion that NO3 dominates over O3 for BVOC losses and for the SOA modeling. The calculation that NO3 accounts for nearly 100% of the BVOC loss at night is a little surprising and also from which major conclusions of this manuscript flow. For the Xu et al. (2015b) study (referenced in the manuscript), they calculated that only 20%/38% of the reacted a/b-binene was with NO3 at night. Might the NO3 calculated here be biased high since only two VOC losses were considered?

**REPLY:**

Since on-line VOCs measurement was only performed during the spring campaign, the following theoretical analysis of $NO_3$+VOCs reactions will be just applied to the spring case. Typical nighttime VOC concentrations, their reaction rate coefficients for reacting with $NO_3$ radical are listed in Table S3. Comparing to five biogenic VOCs (i.e., α-pinene, isoprene, β-pinene, limonene and

camphene )accounting for 99% $NO_3$ loss in Table S3 in Xu et al. (2015b), one anthropogenic VOC, styrene made the third largest contribution to $NO_3$ loss, which should not be ignored in the related SOA estimation analysis. But α-pinene and limonene do contribute to nearly 90% of $NO_3$ loss in our case due to their much higher nighttime concentrations than other BVOCs and rapid reaction rates with $NO_3$ radical. The nighttime estimated concentration of $NO_3$ radical in this study is 1.24±0.76pptv, which is about 15 times higher than the nighttime concentration of $NO_3$ radicals (0.076 pptv) reported in Xu et al. (2015b), this is because that the value of $[NO_2] \times [O_3]$ (20ppbv$\times$6.8ppbv) in this case is just about 15 times higher than that value (0.54ppbv for $NO_2$ and 21 ppbv for $O_3$) in Xu et al. (2015b). Thus, the nighttime concentration of $NO_3$ and $O_3$ is 1.24 pptv and 6.8ppbv, respectively, in this study, while in Xu et al. (2015b), the nighttime concentration of $NO_3$ and $O_3$ is 0.076 pptv and 21ppbv, respectively. The much higher $NO_3$ level and lower $O_3$ level in this study leads to almost all VOCs reacting with $NO_3$ radical over $O_3$ at night.

**Table S3.** The average campaign concentrations of VOCs measured with an automated in situ gas-chromatography mass spectrometer (GC-MS), their reaction rate coefficients for reacting with $NO_3$ radical and the production potential from $NO_3$+VOC in spring.

| VOC species | Mean concentration (ppbv) | Rate Coefficient | Production potential (pptv/s) |
|---|---|---|---|
| 1,2,3-Trimethylbenzene | 0.057 | 1.90E-15 | 3.72E-06 |
| 1,2,4-Trimethylbenzene | 0.177 | 1.80E-15 | 1.10E-05 |
| 1,3,5-Trimethylbenzene | 0.051 | 8.80E-16 | 1.54E-06 |
| 1,3-Butadiene | 0.052 | 1.00E-13 | 1.79E-04 |
| 1-Butene | 0.415 | 1.32E-14 | 1.89E-04 |
| 1-Hexene | 0.022 | 1.20E-14 | 9.06E-06 |
| 1-Pentene | 0.022 | 1.20E-14 | 9.27E-06 |
| 2,2,4-Trimethylpentane | 0.068 | 9.00E-17 | 2.10E-07 |
| 2,2-Dimethylbutane | 0.199 | 4.40E-16 | 3.01E-06 |
| 2,3,4-Trimethylpentane | 0.022 | 1.90E-16 | 1.44E-07 |
| 2,3-Dimethylbutane | 0.299 | 4.40E-16 | 4.54E-06 |
| 2,3-Dimethylpentane | 0.293 | 1.50E-16 | 1.51E-06 |
| 2-Methylheptane | 0.034 | 1.90E-16 | 2.20E-07 |
| 2-Methylhexane | 0.514 | 1.50E-16 | 2.66E-06 |
| 2-Methylpentane | 1.582 | 1.80E-16 | 9.81E-06 |
| 3-Methylheptane | 0.027 | 1.90E-16 | 1.74E-07 |
| 3-Methylhexane | 0.534 | 1.50E-16 | 2.76E-06 |
| 3-Methylpentane | 1.411 | 2.20E-16 | 1.07E-05 |
| Acetaldehyde | 1.249 | 2.70E-15 | 1.16E-04 |
| Acetylene | 0.941 | 5.10E-17 | 1.65E-06 |
| Acrolein | 0.042 | 3.30E-15 | 4.73E-06 |
| Benzene | 0.599 | 3.00E-17 | 6.19E-07 |
| cis-2-Pentene | 0.005 | 3.70E-13 | 6.57E-05 |
| Cyclohexane | 1.164 | 1.40E-16 | 5.61E-06 |
| Cyclopentane | 0.416 | 1.40E-16 | 2.00E-06 |
| Ethane | 1.567 | 1.00E-17 | 5.40E-07 |

| | | | |
|---|---|---|---|
| **Ethylbenzene** | 0.563 | 1.20E-16 | 2.34E-06 |
| **Isoprene** | 0.032 | 6.96E-13 | 7.76E-04 |
| **m/p-Xylene** | 0.602 | 3.80E-16 | 7.88E-06 |
| **Methacrolein** | 0.012 | 3.40E-15 | 1.44E-06 |
| **Methylcyclohexane** | 0.172 | 1.40E-16 | 8.29E-07 |
| **Methylcyclopentane** | 0.673 | 1.40E-16 | 3.25E-06 |
| **n-Butanal** | 0.044 | 1.10E-14 | 1.68E-05 |
| **n-Butane** | 1.848 | 4.60E-17 | 2.93E-06 |
| **n-Decane** | 0.060 | 2.80E-16 | 5.74E-07 |
| **n-Heptane** | 0.351 | 1.50E-16 | 1.81E-06 |
| **n-Hexane** | 1.916 | 1.10E-16 | 7.25E-06 |
| **n-Nonane** | 0.033 | 2.30E-16 | 2.59E-07 |
| **n-Pentanal** | 0.128 | 1.50E-14 | 6.61E-05 |
| **n-Pentane** | 0.593 | 8.70E-17 | 1.78E-06 |
| **n-Propylbenzene** | 0.029 | 6.00E-16 | 6.01E-07 |
| **Octane** | 0.064 | 1.90E-16 | 4.17E-07 |
| **o-Xylene** | 0.464 | 3.80E-16 | 6.06E-06 |
| **Propanal** | 0.144 | 6.31E-15 | 3.12E-05 |
| **Propane** | 3.678 | 7.00E-17 | 8.86E-06 |
| **Propene** | 0.477 | 9.54E-15 | 1.57E-04 |
| **Styrene** | 0.194 | 1.50E-12 | 1.00E-02 |
| **Toluene** | 3.120 | 7.00E-17 | 7.52E-06 |
| **alpha-Piene** | 0.391 | 6.21E-12 | 8.36E-02 |
| **beta-Piene** | 0.013 | 2.51E-12 | 1.10E-03 |
| **Camphene** | 0.276 | 6.20E-13 | 5.91E-03 |
| **Limonene** | 0.137 | 1.22E-11 | 5.77E-02 |

6. As Referee 2 points out regarding the analysis in Sect. 3.3 on the estimation of pON formation, the sources of measured pON depend on reacted VOCs, not on the amount of VOCs present in the atmosphere. The authors seem to be equating the two. Essentially the authors appear to be calculating the relative amounts of production expected if the sampled airmass was allowed to react to completion with no further emissions. This does not equate to the regional productions since it would systematically underweight more reactive compounds and the two metrics may only be loosely connected. In general, Section 3.3. is highly undersupported and speculative considering the evidence shown. It is not clear what the modeled SOA (blue trace in Fig. 5) even is. Is that the amount produced per unit time from the model? It cannot be the cumulative production since it increases and decreases (and the model does not have dynamics, dilution, mixing, etc.). In any case, the features of organic nitrates and the ambiguous modeled SOA don't match all that well. Also, it seems likely that the day-to-day variability may be of similar or larger magnitude to the variations in the average diurnal cycle (variability bars such as standard deviations would be helpful here). Potentially the very rough similarities may be an averaging artifact? Thus, the diurnal cycle should be supplemented additional evidence, such as time series of the relevant metrics and correlations plots to make a convincing case that the model may be indeed be representing the key processes and explain the measurements. Also, inclusion of inorganic nitrates together with the organic nitrates (diurnal cycle and other relevant plots), would help make a stronger case that the

separation of the organic nitrates is meaningful and robust. Based on this section, the authors conclude (as stated in the abstract) that BVOC + NO3 at night are the dominant formation pathway of organic nitrates in the polluted atmosphere. This simply has not been demonstrated. Moreover, it appears that the authors are equating boundary layer concentration with overall regional production importance. As the authors note in Sect. 3.3, the boundary layer is expected to be lower during nighttime. Consequently, the concentrations observed represent a smaller volume of air, so equating lower concentrations during daytime with lower overall (column integrated, regional) importance is faulty logic. BL-effects were not considered here nor production during the daytime modeled, thus no conclusions beyond nighttime boundary layer concentrations and production should be drawn based on this analysis. Yet, this manuscript seems to do just that – an example of the vastly overstated implications claimed.

**REPLY:**

We have accepted this comment and changed the analysis in section 3.4 significantly. First, we used the NO$_3$ loss rate at night, which can be calculated as $K_i \cdot [VOC_i]$ in Eq. (9), to roughly judge the production potential of organic nitrates from a NO$_3$+VOC reaction:

$$[\text{Production Potential}]_{NO3+VOCi} = K_i \cdot [VOC_i] \cdot [NO_3] \quad (9)$$

Where $K_i$ represents the reaction rate coefficient for NO$_3$ radical and a VOC, $[VOC_i]$ is the concentration of the specific VOC and $[NO_3]$ is the concentration of NO$_3$ radical. In the spring campaign, the diurnal variations of NO$_2$, O$_3$ and estimated NO$_3$ radical concentrations are shown in Figure S10 (with standard deviations). It was found that the high concentrations of NO$_2$ (19.93±2.31 ppb) at night leaded to high yield of NO$_3$ radical (1.24±0.76 ppt) in Shenzhen.

According to the distribution of production potential, five biogenic VOCs (BVOCs) (i.e., α-pinene, limonene, camphene, β-pinene and isoprene) and one anthropogenic VOC (styrene) were identified as notable VOC precursors with high production potential, while the sum of production potential from the other VOCs was negligible as shown in Figure 5b.

In addition, Figure 6 shows the average nighttime variations of BC, LO-OOA, NO$_{3.org\_ratio\_1}$, NO$_{3.org\_PMF}$ and production potential of the six notable VOCs identified during the spring campaign. The concentrations of BC and LO-OOA generally decreased slowly after sunset till sunshine due to the combined effect of both the planetary boundary layer variation and traffic emissions, while particulate organic nitrates showed a different trend with two clear growth processes (19:00-22:00 and 3:00-6:00) at night, suggesting their unique sources. In contrast, the production potentials of the six notable VOCs with NO$_3$ had two roughly similar increases at the same periods as those of particulate organic nitrates, which supported the key role of NO$_3$+VOCs reactions for nighttime organic nitrate formation.

Finally, based on the production potential evaluation above, we further estimated roughly the nighttime SOA bulk yield of NO$_3$+the six notable VOC precursors. And the estimated SOA production from NO$_3$+VOCs reactions using SOA mass yields in the literature was 0-0.33 μg m$^{-3}$ for α-pinene, 0.09-1.28 μg m$^{-3}$ for limonene, 0.24 μg m$^{-3}$ for styrene, 0.004-0.06 μg m$^{-3}$ for β-pinene and 0.002-0.02 μg m$^{-3}$ for isoprene. The SOA yield from camphene is currently unknown in the literature. It is seen that the average observed nighttime concentration of particulate organic nitrates during the spring campaign (0.39-0.83μg m$^{-3}$, converting NO$_{3.org\_ratio\_1}$, NO$_{3.org\_PMF}$ in Figure 6 into organic nitrates assuming the average molecular weight of organic nitrates of 200 to 300 g mol$^{-1}$) was well within the estimated SOA concentration ranges produced by α-pinene, limonene and styrene, indicating that these three VOCs were the key VOC precursors in urban atmosphere in Shenzhen. Considering both the production potentials and SOA yields, the contributions of β-pinene and isoprene to nighttime formation of particulate organic nitrates could be negligible. Besides the BVOCs species, this study

highlights the key role of anthropogenic styrene in nighttime particulate organic nitrate formation in urban atmosphere in China, and relevant smog chamber studies for anthropogenic VOCs+NO$_3$ reactions are needed to support parameterization in modeling.

7. Kiendler Scharr et al. (2016), Ng et al. (2017), Xu et al. (2015a, 2015b) and others have reported that at many sites throughout the US and Europe, including polluted urban areas, organic nitrates can be substantial fractions of aerosol nitrates and NO3+BVOC reactions can be an important source. The manuscript fails to make a case for what new information this study from one polluted city provides and how it would add to the body of atmospheric literature.

**REPLY:**

In the revised manuscript, we have proposed several new information: 1.adding some description in *Introduction* to address the contribution of this paper on organic nitrates studies in detail. "Ng et al. (2017) reviewed the nitrate radical chemistry and the abundance of particulate organic nitrates in the United States and Europe, and further concluded that particulate organic nitrates are formed substantially via NO$_3$+BVOC chemistry, which plays an important role in SOA formation. Unfortunately, relevant Chinese datasets are scarce yet and not included in this review. This was because (1) the contributions of organic nitrates in SOA and total nitrates in Chinese atmosphere remain poorly understood; (2) the anthropogenic and biogenic precursor emissions in China are largely different from those in the United States and Europe, and thus cannot be easily estimated. To our best knowledge, few studies have investigated the concentrations and formation pathways of particulate organic nitrates in China. Xu et al. (2017) estimated the mass concentration of organic nitrogen in Beijing using AMS, but in this study they ignored the contribution of NO$_X^+$ family, which are the major fragments of organic nitrates"; 2. in section 3.3, we discussed the size distribution characteristics of organic nitrates. We used the NO$^+$/NO$_2^+$ ratio as an indicator to investigate the size distribution characteristics of organic nitrates. It is clearly found that the NO$^+$/NO$_2^+$ ratio generally increases towards smaller size in spring, summer, and autumn, while the NO$^+$/NO$_2^+$ ratio keep similar to the value of R$_{NH4NO3}$ throughout the full size range in winter. It should also be noted that in spring, summer, and autumn, the lowest values of NO$^+$/NO$_2^+$ ratio at > 1 μm are also approximate to the seasonal values of R$_{NH4NO3}$. These characteristics clearly indicate that organic nitrates occurred mostly in fresh particles with smaller sizes, and thus should be mainly of local origin. The diurnal trends of size distributions of NO$^+$/NO$_2^+$ ratio show higher values in small size range at night comparing that in the day in spring, summer and autumn, suggesting a dominant nighttime origin of organic nitrates; 3. according to the analysis in section 3.4, we can find that besides the BVOCs species, anthropogenic styrene plays an important role in nighttime particulate organic nitrate formation in urban atmosphere in China; 4. we compared this study with other particulate organic nitrate studies in section 3.5 and the results show that the formation of particulate organic nitrate is more likely NOx-control than BVOCs-control and high NOx emissions could promote biogenic SOA formation at night.

8. The authors broadly refer to their results as pertaining to "South China" which is a large and diverse areas. Unless there is evidence that this site is generally representative of that geographic area, text and conclusions should be limited to just this one urban area. This is an example of overselling the story without providing the supporting analysis.

**REPLY:**

We have toned down this conclusion and only addressed this case as a typical urban site in South China.

9. L32-3: "Play a larger role" for what? Reference?

**REPLY:**

We have added the related references.

10. L40-41: The Rollins et al. (2002) paper demonstrated the application of the aerosol-only organic nitrate measurement. The technique for total nitrates was developed and demonstrated a decade earlier (Day et al., 2002).

**REPLY:**

We have corrected it.

11. L41: "measured" would be better than "obtained"

**REPLY:**

We have corrected it.

12. L66: "to obtain more representative samples" seems vague. Clarify.

**REPLY:**

We have deleted "more representative".

13. L74: "literatures" should not be plural.

**REPLY:**

We have corrected it.

14. L82-4: state if the RIE of ammonium was calibrated or assumed.

**REPLY:**

RIE of ammonium was assumed and we have added the statement in section 2.2.1: The relative ionization efficiencies (RIEs) used in the study were 1.2 for sulfate, 1.1 for nitrate, 1.3 for chloride, 1.4 for organics, and 4.0 for ammonium (Jimenez et al., 2003).

15. L105-6: Site Xu et al. (2015a) since isn't this is exactly what they did?

**REPLY:**

We have added this reference.

16. L107-8: Stating that negative calculated organic nitrates means that concentrations must be low is not very analytically sound. E.g., can't this just mean the method isn't working well or there are large uncertainties due a variety of possible factors? Please revise to be more precise and inclusive of the possible causes.

**REPLY:**

We have modified this statement and provided more detailed and precise discussion in section 3.1.

17. L127: Fry et al. 2013 is not the proper reference for heterogeneous N2O5 reaction with aerosol. They just performed the calculation.

**REPLY:**

We have replaced the right literature (Dentener and Crutzen, 1993) with it in the manuscript.

18. L130: Units for velocity are wrong.

**REPLY:**

We have corrected it.

19. L131: Where did the aerosol surface area concentration for the modeling come from? It does not appear that an aerosol sizing instrument was used in the study.

**REPLY:**

SA is calculated from the size-resolved particle number concentrations assuming spherical particles measured by a scanning mobility particle sizer (SMPS) (TSI Inc., USA, 3775 CPC and TSI Inc. 3080 DMA). And 220 $\mu m^2 cm^{-3}$ is actually under dry conditions, the ambient (wet) aerosol SA is 475 $\mu m^2 cm^{-3}$ by using the hygroscopic growth factor in Liu et al. (2010). We have added and corrected the related description in section 2.2.2 and Text S1.

20. L133-135: Why do the authors include the calculation of the N2O5+H2O gas-phase reaction? The lifetime is 500 years!

**REPLY:**

We have corrected the value of daily maximum $[H_2O]$ ($5.5*10^{17}$ molecule $cm^{-3}$) and the calculated value of $N_2O_5$ lifetime with respect to the reaction with $H_2O$ (1470 s).

21. L145: "6.82 and 19.38 ppb": Too many significant figures. It would be more useful to report averages and standard deviations.

**REPLY:**

The concentrations of $NO_2$ and $O_3$ with their standard deviations are shown in Figure S10 in the supplementary.

22. L159: "may explain" instead? Without any quantitative assessment, it is not justified to say that it does explain it.

**REPLY:**

We have replace "explain" with "may explain".

23. L165: "adequate" for what or by what measure? Too vague – needs clarification.

**REPLY:**

We have replace "adequate" with "good".

24. L173-4: Statement points out that the organic nitrate is similar to SE US "even though" BC and NOx are higher in S. China. How is this meaningful? If terpenes are dominantly reacting with NO3 then the production would be largely controlled by the amount of terpenes present - which appear to be quite modest at this location. Statements like this need more context/discussion to be meaningful. Or remove such comparisons if not informative and making a clear point.

**REPLY:**

We have deleted this statement.

25. L180: "both organic spectra" should instead read "both OOA spectra"?

**REPLY:**

We have corrected it

26. L180-2: Unclear. Is this referring to running PMF with and without the NOx ions? Please clarify.
**REPLY:**
We have clarified this statement: The mass spectrum profiles and diurnal patterns of each OA factor using PMF based on OA spectra only in spring, summer and autumn are shown in Figure S2-S5.

27. 182-3: Please report these correlations (table?) and show the correlations in the SI (i.e. duplicate Fig. 3 for MO-OOA and HOA).
**REPLY:**
We have added Figure S7-S9 to show the correlations between organic nitrates with HOA and MO-OOA.

28. L196-198: Limonene and a-pinene shown to account for 90% of NO3 loss? Assuming the authors are referring to Sect. 2.4 (not 2.3 as written), this is not shown there – or anywhere else. Only 2 compounds were considered according to the text (which references Table S1).
**REPLY:**
We have added the related information in section 3.4 and table S3.

29. Table 1: "NO3-": Is that inorganic nitrate or total nitrate? If total, then the ionic denotation isn't appropriate.
**REPLY:**
We have replace "$NO_3^-$" with "total $NO_3^-$".

30. Figure 5: beta-pinene misspelled.
**REPLY:**
We have corrected it.

31.Table S1: a-pinene SOA yield: Why was this single reference value picked, considering that there is a substantial range reported in the literature? See Table 2 in Ng et al. (2017). A range would seem more appropriate here unless the authors justify why this particular one is more appropriate for this study.
**REPLY:**
We have added more references of SOA yield, please see Table 3 in the manuscript.

**References**

Liu, X.G., Zhang, Y.H., Wen, M.T., Wang, J.L., Jung, J.S., Chang, S.-Y., Hu, M., Zeng, L.M. and Kim,

    Y.J.: A closure study of aerosol hygroscopic growth factor during the 2006 Pearl River Delta

    Campaign. Adv. Atmos. Sci. 27, 947–956, 2010.

---

## Editor Decision (ED1)

The authors have made substantial improvements addressing the referrees' concerns, by presenting additional supporting plots and explanations, putting the measurements more in context of other studies, and reframing the implications and conclusions to better reflect the evidence presented. However, there still appear to be some explanations and conclusions drawn that are not well supported – and therefore should not be published in current form. Details are below along with a few other suggestions/questions.

Section 3.4 and Fig. 6 on model/measurement comparisons:

It is stated "The concentrations of BC and LO-OOA generally decreased slowly after sunset till sunshine due to the combined effect of both the planetary boundary layer variation and traffic emissions while particulate organic nitrates showed a different trend with two clear growth processes (19:00-22:00 and 3:00-6:00) at night, suggesting their unique sources. In contrast, the production potentials of the six notable VOCs with NO3 had two roughly similar increases at the same periods as those of particulate organic nitrates, which supported the key role of NO3+VOCs reactions for nighttime organic nitrate formation."

How do PBL and traffic emissions explain the steady decline of BC and LO-OOA? More clarification is needed on why this would be the case and how it is known. Generally, statements like this seem to be made a bit haphazardly throughout the manuscript – i.e. strongly stating an explanation for something that is likely a complex process, while providing little explanation/evidence. Loose language/science like this is detrimental to the scientific literature.

More importantly, the statement that the nocturnal cycles of the PP and organic nitrate match is simply not supported by the data! It seems to rather describe what the authors hoped to see in the data based on the modeling. The PP for the dominant SOA-forming compounds shows a 2-4 fold increase at 21-1 hrs while the organic nitrates show subtle decreases. Thus, the observed aerosol data does not seem to support any connection between modeled sources and measurements based on what is shown. The authors should revise the discussion accordingly to accurately reflect what can be supported with the data. As is, it seems to suggest that either the modeling framework does not represent the dominant chemistry producing organic nitrates and SOA or there are other factors at play that make the effects difficult to observe. Perhaps there are other more important anthropogenic precursors, that were not measured, that build up during the nighttime in the lower nocturnal BL and react with NO3 or O3? That would be consistent with the stronger correlations with NO2 than with the VOCs show in Fig. S11.

As I suggested in my initial review, the subtle features in the organic nitrates may be just driven by averaging artifacts and further investigation of time series or correlations may shed some light on the connections (or lack thereof). For example, during nights when the PP is particularly high, are the organic nitrates and LO-OOA correspondingly elevated? How do the correlations of LO-OOA and organic nitrates vs calculated PP look? Or better yet, versus computed particle organic nitrate or SOA production. Without further compelling observational evidence, any conclusions about the importance of particular VOCs+NO3 sources should be strictly framed as theoretical, based on the modeling of potential sources throughout the paper. In fact, in that case, the possibility that other anthropogenic VOC may be driving the nighttime organic nitrate production should be considered and discussed.

Also, if the authors choose to keep Fig. 6, it would be very useful to include the nocturnal cycle of the calculated SOA and organic nitrate formation rate since that is more directly related to the particle-phase measurements being discussed.

Sect. 3.3 on chemically-resolved size distributions:

This an interesting addition since the original manuscript. To my knowledge high-resolution nitrate and/or NO+ and NO2+ has not been reported before in the published literature. The authors should confirm that the NO+/NO2+ ratio was computed using high-resolution PToF analysis (since UMR m/z 30/46 would be much less meaningful due to CH2Ox+ interferences). Therefore, can the authors please show size distributions of the NO+ and NO2+ signals for the different seasons (in SI)? This would help to understand how well the HR PToF worked in separating those two ions and provide a helpful example for future analyses using AMS HR-PToF data.

Additionally, if this is indeed HR-PToF, I would strongly encourage that the authors compute the size distributions of organic and inorganic nitrate separately (using the NOx+ ratio equations and the total nitrate, NO+ and NO2+ size distributions) and show them in Figure 4. That would allow more direct and intuitive inspection of where the different modes reside (particularly for the non-AMS aficionado). And relatedly, showing that organic nitrates size distributions are indeed significantly different from the OA distributions is critical to support the interpretation in this section that the size-dependence of the NOx ratio indicates fresher SOA sources.

Section 3.5 on comparison to other studies:

It is stated "Note that, particulate organic nitrates constituted the major part (86-100%) of total nitrates in the atmosphere scarce of NOx (in Centreville and Woodland Park), suggesting that NOx was very quickly consumed to form particulate organic nitrates and thus the formation of particulate organic nitrates should be NOx-limited."

This is wild speculation and almost certainly false. The organic nitrate concentrations at those sites were much lower than NOx and simply the fact that organic nitrate >> inorganic nitrate does not provide a direct line of reasoning to what the major losses of NOx are. Lower NOx at those sites was likely due to the fact they were removed from strong urban emissions. Moreever, the use of "NOx-limited" here and throughout the paper is loosely and ambiguously invoked. With lower NOx concentrations, NO3 radical production may decrease, however the BVOC will still most certainly instead react with O3 or OH and form organic nitrates and SOA. In this section and throughout the authors need to clarify what is meant by NOx-limited – including discussion of other literature for context. As is, all discussions of "NOx-limited" is ambiguous and underdeveloped. I think for the analysis of this particular study being analyzed, the authors mean that it is driven by NO3 chemistry. But that is not the same as "NOx-limited".

Table 2 and caption and manuscript text: variability with FPEAK is not a metric of "uncertainty". Bootstrapping is typically used as an estimate of uncertainties. "Uncertainties" in this context should be replaced with something like "variability". Also, it is not clear what the number in the table reported is. Is that the stand deviation of all values with different FPEAK, or the max difference from the FPEAK=0 solution? Also it should be written as FPEAK, not fpeak.

Figure 6: "alpha" is misspelled.

Table S3. Alpha and beta Pinene are misspelled.

General: There are a quite a few grammatical mistakes, particularly in the new text that was added.

---

## Author Response (AR2)

Report #1

1. In Figure 4(a), please add the variation of NO+/NO2+ ratio as a function of diameter.

**REPLY:**

We have added it in Figure 4(a).

2. In Eqn. (9), the organic nitrate yield should be included to calculate "production potential of organic nitrates". $K_i \cdot [VOC_i] \cdot [NO_3]$ only represents the loss rate of NO3 radical.

**REPLY:**

Yes, the organic nitrate yield should be included to calculate "production potential of organic nitrates" in Eqn. (9). However, we didn't multiply the organic nitrates yield parameters in calculative process, because only a few organic nitrate yields for BVOCs were available in the literature (Fry et al., 2014; Ng et al., 2017). However, given the fact that the values of $K_i \cdot [VOC_i] \cdot [NO_3]$ for different VOC species can differ by orders of magnitude, not multiplying the organic nitrate yields (ranging from 0 to 1) would not significantly affect the PP ranking of VOCs. We have added the detailed explanation in section 3.4.

3. When using Eqn. (11), I suggest to use "potential SOA formation" in the discussion.

**REPLY:**

We have amended it in related discussion.

**References**

Fry, J.L., Draper, D.C., Barsanti, K.C., Smith, J.N., Ortega, J., Winkler, P.M., Lawler, M.J., Brown, S.S., Edwards, P.M., Cohen, R.C.: Secondary Organic Aerosol Formation and Organic Nitrate Yield from NO3 Oxidation of Biogenic Hydrocarbons. Environ. Sci. Technol. 48, 11944–11953. https://doi.org/10.1021/es502204x, 2014.

Ng, N. L., Brown, S. S., Archibald, A. T., Atlas, E., Cohen, R. C., Crowley, J. N., Day, D. A., Donahue, N. M., Fry, J. L., Fuchs, H., Griffin, R. J., Guzman, M. I., Herrmann, H., Hodzic, A., Iinuma, Y., Jimenez, J. L., Kiendler-Scharr, A., Lee, B. H., Luecken, D. J., Mao, J., McLaren, R., Mutzel, A., Osthoff, H. D., Ouyang, B., Picquet-Varrault, B., Platt, U., Pye, H. O. T., Rudich, Y., Schwantes, R. H., Shiraiwa, M., Stutz, J., Thornton, J. A., Tilgner, A., Williams, B. J., and Zaveri, R. A.: Nitrate radicals and biogenic volatile organic compounds: oxidation, mechanisms, and organic aerosol, Atmos. Chem. Phys., 17, 2103-2162, https://doi.org/10.5194/acp-17-2103-2017, 2017.

Report #2

The authors have made substantial improvements addressing the referrees' concerns, by presenting additional supporting plots and explanations, putting the measurements more in context of other studies, and reframing the implications and conclusions to better reflect the evidence presented. However, there still appear to be some explanations and conclusions drawn that are not well supported – and therefore should not be published in current form. Details are below along with a few other suggestions/questions.

1. Section 3.4 and Fig. 6 on model/measurement comparisons:

It is stated "The concentrations of BC and LO-OOA generally decreased slowly after sunset till sunshine due to the combined effect of both the planetary boundary layer variation and traffic emissions while particulate organic nitrates showed a different trend with two clear growth processes (19:00-22:00 and 3:00-6:00) at night, suggesting their unique sources. In contrast, the production potentials of the six notable VOCs with NO3 had two roughly similar increases at the same periods as those of particulate organic nitrates, which supported the key role of NO3+VOCs reactions for nighttime organic nitrate formation."

How do PBL and traffic emissions explain the steady decline of BC and LO-OOA? More clarification is needed on why this would be the case and how it is known. Generally, statements like this seem to be made a bit haphazardly throughout the manuscript – i.e. strongly stating an explanation for something that is likely a complex process, while providing little explanation/evidence. Loose language/science like this is detrimental to the scientific literature.

More importantly, the statement that the nocturnal cycles of the PP and organic nitrate match is simply not supported by the data! It seems to rather describe what the authors hoped to see in the data based on the modeling. The PP for the dominant SOA-forming compounds shows a 2-4 fold increase at 21-1 hrs while the organic nitrates show subtle decreases. Thus, the observed aerosol data does not seem to support any connection between modeled sources and measurements based on what is shown. The authors should revise the discussion accordingly to accurately reflect what can be supported with the data. As is, it seems to suggest that either the modeling framework does not represent the dominant chemistry producing organic nitrates and SOA or there are other factors at play that make the effects difficult to observe. Perhaps there are other more important anthropogenic precursors, that were not measured, that build up during the nighttime in the lower nocturnal BL and react with NO3 or O3? That would be consistent with the stronger correlations with NO2 than with the VOCs show in Fig. S11.

As I suggested in my initial review, the subtle features in the organic nitrates may be just driven by averaging artifacts and further investigation of time series or correlations may shed some light on the connections (or lack thereof). For example, during nights when the PP is particularly high, are the organic nitrates and LO-OOA correspondingly elevated? How do the correlations of LO-OOA and organic nitrates vs calculated PP look? Or better yet, versus computed particle organic nitrate or SOA production. Without further compelling observational evidence, any conclusions about the importance of particular VOCs+NO3 sources should be strictly framed as theoretical, based on the modeling of potential sources throughout the paper. In fact, in that case, the possibility that other anthropogenic VOC may be driving the nighttime organic nitrate production should be considered and discussed.

Also, if the authors choose to keep Fig. 6, it would be very useful to include the nocturnal cycle of the calculated SOA and organic nitrate formation rate since that is more directly related to the particle-phase measurements being discussed.

We thank the reviewer for the valuable advice. We have reorganized the statement for Figure 6 in section 3.4 and provided more evidence to support the correlation between $NO_3$+VOCs reactions and the organic nitrate formation. Figure 6 also has been revised to diurnal cycle instead of nocturnal cycle to see clearly the variations between day and night. Unfortunately, we can't add the cycles of the calculate SOA and organic nitrates formation rate into Figure 6 because the related kinetic reaction parameters and other important factors (i.e., dynamics, dilution, mixing, etc.) are not available. However, we have done other important investigation as the reviewer suggested, including the correlations between PPs with organic nitrates and LO-OOA and addressing the possible contribution of other important anthropogenic VOC precursors that would be responsible for the formation of organic nitrates but has not yet been identified. The revised statement in section 3.4 is followed as:

"The estimation of potential SOA production above suggests significant contributions of α-pinene, limonene, and styrene to SOA, and the significant contribution of camphene is also possible. Thus, we further explored the diurnal variations of the PPs of these four VOCs. Figure 6 shows the diurnal trends of BC, LO-OOA, $NO_{3.org\_ratio\_1}$, $NO_{3.org\_PMF}$, and the PPs of the four VOCs during the spring campaign. There were two apparent nighttime growth periods (i.e., I: 19:00–22:00 and II: 2:00–6:00) for both $NO_{3.org\_ratio\_1}$ and $NO_{3.org\_PMF}$. During Period I, BC maintained a relatively higher level, suggesting stable anthropogenic emissions. In contrast, the increases of all the PPs during Period I indicated that these precursors contributed to the organic nitrate growth. After 22:00, while the PPs still showed a rapid growth, BC and organic nitrates began to decrease, implying possible existence of other important anthropogenic VOC precursors, which were not identified by the GC-FID/MS analysis but would dominate the formation of organic nitrates at this stage. During Period II, the anthropogenic emissions remained at a stable lower level, as indicated by BC, while all the PPs increased with organic nitrates again, indicating that these four precursors also contributed to, or could dominate, this organic nitrate growth. As shown in Figure S13, organic nitrates correlated better with the PPs (R=0.63–0.74) than with LO-OOA (R=0.19–0.31) or BC (R=0.02-0.05) during Period II at the spring campaign, suggesting the significant contributions of the $NO_3$ reactions with these precursors."

2. Sect. 3.3 on chemically-resolved size distributions:

This an interesting addition since the original manuscript. To my knowledge high-resolution nitrate and/or NO+ and NO2+ has not been reported before in the published literature. The authors should confirm that the NO+/NO2+ ratio was computed using high-resolution PToF analysis (since UMR m/z 30/46 would be much less meaningful due to CH2Ox+ interferences). Therefore, can the authors please show size distributions of the NO+ and NO2+ signals for the different seasons (in SI)? This would help to understand how well the HR PToF worked in separating those two ions and provide a helpful example for future analyses using AMS HR-PToF data.

Additionally, if this is indeed HR-PToF, I would strongly encourage that the authors compute the size distributions of organic and inorganic nitrate separately (using the NOx+ ratio equations and the total nitrate, $NO^+$ and $NO_2^+$ size distributions) and show them in Figure 4. That would allow more direct and intuitive inspection of where the different modes reside (particularly for the non-AMS aficionado). And relatedly, showing that organic nitrates size distributions are indeed significantly different from the OA distributions is critical to support the interpretation in this section that the size-dependence of the NOx ratio indicates fresher SOA sources.

**REPLY:**

Due to the lack of HR-PToF data, our analyses used the UMR-PToF data (m/z 30 and 46), which may bring in the interferences of $CH_2O_X^+$. However, for all four seasons, the contributions of $CH_2O_X^+$ in m/z 30 and 46 in the HR data of $PM_1$ were less than 10% (Figure S10), which indicates that the interferences were negligible in this study. The size distributions of the $NO^+$ and $NO_2^+$ signals for the different seasons have been shown in Figure S11.

3. Section 3.5 on comparison to other studies:

It is stated "Note that, particulate organic nitrates constituted the major part (86-100%) of total nitrates in the atmosphere scarce of NOx (in Centreville and Woodland Park), suggesting that NOx was very quickly consumed to form particulate organic nitrates and thus the formation of particulate organic nitrates should be NOx-limited."

This is wild speculation and almost certainly false. The organic nitrate concentrations at those sites were much lower than NOx and simply the fact that organic nitrate >> inorganic nitrate does not provide a direct line of reasoning to what the major losses of NOx are. Lower NOx at those sites was likely due to the fact they were removed from strong urban emissions. Moreover, the use of "NOx-limited" here and throughout the paper is loosely and ambiguously invoked. With lower NOx concentrations, NO3 radical production may decrease, however the BVOC will still most certainly instead react with O3 or OH and form organic nitrates and SOA. In this section and throughout the authors need to clarify what is meant by NOx-limited – including discussion of other literature for context. As is, all discussions of "NOx-limited" is ambiguous and underdeveloped. I think for the analysis of this particular study being analyzed, the authors mean that it is driven by NO3 chemistry. But that is not the same as "NOx-limited".

**REPLY:**

We have amended the ambiguous and undeveloped statement in section 3.5. The relevant changes are as follow:

"Higher concentrations of particulate organic nitrates generally is associate with higher NOx concentrations rather than BVOC concentrations. On the other hand, although the BVOC concentrations in Bakersfield were far less than that in other campaigns, the concentration of particulate organic nitrates there showed an intermediate level among all the campaigns. Therefore, it is suggested that the formation of particulate organic nitrates may be more relevant with NOx than BVOCs, which is consistent with the finding that the organic nitrate production was dominated by NOx in the southeastern US (Edwards et al., 2017). In the spring campaign of this study, we examined the correlation between organic nitrates and $NO_2$ or VOCs (by the sum of α-pinene, limonene and styrene) at night (Figure S14) and found a significant correlation of organic nitrates with $NO_2$ (R=0.40-0.47) rather than with VOCs (R=0.22-0.23), which further suggests that the organic nitrates formation was driven by the NOx-involved $NO_3$ chemistry."

4. Table 2 and caption and manuscript text: variability with FPEAK is not a metric of "uncertainty". Bootstrapping is typically used as an estimate of uncertainties. "Uncertainties" in this context should be replaced with something like "variability". Also, it is not clear what the number in the table reported is. Is that the stand deviation of all values with different FPEAK, or the max difference from the FPEAK=0 solution? Also it should be written as FPEAK, not fpeak.

**REPLY:**

We have changed "Uncertainties" to "standard deviations" and "fpeak" to "FPEAK" in the context.

5. Figure 6: "alpha" is misspelled.

**REPLY:**

We have corrected it.

6. Table S3. Alpha and beta Pinene are misspelled.

**REPLY:**

We have corrected it.

7. General: There are a quite a few grammatical mistakes, particularly in the new text that was added.

**REPLY:**

We have amended them.

[revised manuscript text omitted]

---

## Author Response (AR3)

1. Regarding the discussions of the size distributions of NO+ and NO2+ ions. It is good the authors have clarified that this is in fact UMR m/z 30 and m/z 46. Consequently, all denotations of those should be changed to m/z 30 and m/z 46 as NOx+ ions should only be used to indicate HR-resolved ions.

However, the discussions/conclusions regarding those size distributions are still a bit under-supported/overstated. While it is stated that <10% of those ions were from CH2Ox+ ions, on average for the HR-MS data, there are 3 reasons that does not suffice along to then go on and make a strong argument that the size-dependent trends in m/z30 / m/z46 demonstrated organic nitrates dominate at small sizes and are not present at large sizes:

1) The bulk MS data may be dominated by concentrations at larger sizes, thus the CH2O+ ions may in fact comprise much larger fractions than 10% compared to either NOx+ ion at the smaller sizes. In fact, for biogenic SOA, it would be surprising if that were not the case. See published spectra of biogenic SOA. For example, in a comparison of UMR and HR nitrate quantification, Fry et al. [2018] showed that on average for a biogenically-influenced region, CH2O+ was half of the signal at m/z 30. The author could estimate some rough upper limits that the presence of interfering ions could have on the actual NOx ratio at lower sizes. At minimum, this aspect should be acknowledged.

2) The average contributions shown in Fig. S10 as HR spectra, are season averages. It is not clear to a reader if they are representative of the nighttime periods of interest. An analysis using the time-dependent relative contributions could have been more useful.

3) Just because the contribution to nitrate at lower sizes appears to be more from organic nitrates, does not mean they are not present at larger sizes. There seems to be some faulty logic there. This was the reason I suggested in the past review to calculate size distributions of organic and inorganic rather than only framing the discussions in terms of the inorganic/organic relative apportionment which is less relevant to the scientific arguments being made in this paper about organic nitrate production. Maybe the inorganic nitrate is just less present at smaller sizes.

**REPLY:**

We have changed all relevant $NO^+/NO_2^+$ to m/z 30/m/z 46 in section 3.3. And the time variations of contributions of $CH_2O^+$ in m/z 30 and $CH_2O_2^+$ in m/z 46 in the HR data of $PM_1$ for the four seasons are shown in Figure S10. We further checked the size distributions of m/z 30/m/z 46 ratio under the highest (>15%) and lowest interferences (<5%) $CH_2O_X^+$ interferences in spring, summer and autumn (Figure S12), the results show that m/z 30/m/z 46 variation patterns are not significantly affected by the interferences of $CH_2O_X^+$. Additionally, we used the size distributions of the m/z 30/m/z 46 ratio to separate the size distributions of inorganic and organic nitrates, as shown in Figure S13 according to the suggestion. The statements about organic nitrates are not present at larger sizes are deleted and we only address that organic nitrates were relatively more concentrated at small sizes compared to inorganic nitrates. The relevant changes in section 3.3 are as follow:

"In this section, we attempt to use the $NO^+/NO_2^+$ ratio as an indicator to investigate the size distribution characteristics of organic nitrates. Unfortunately, due to the lack of HR-PToF data, our analyses used the UMR-PToF data of m/z 30 and 46, which might contain the interferences of $CH_2O_X^+$ (Fry et al., 2018). In our case, the time variations of contributions of $CH_2O^+$ in m/z 30 and $CH_2O_2^+$ in m/z 46 in the HR data of $PM_1$ for the four seasons are shown in Figure S10. For all the four seasons, the average contributions of $CH_2O_X^+$ in m/z 30 and 46 in the HR data of $PM_1$ were less than 10%, suggesting that the m/z 30/m/z 46 ratio could mostly represent the $NO^+/NO_2^+$ ratio. The average size distributions of m/z 30 and m/z 46 for the four seasons are shown in Figure S11, and Figure 4a shows the average size distributions of different aerosol species and the m/z 30/m/z 46 ratio in the four seasons. It is clearly found that the m/z 30/m/z 46 ratio exhibited a decreasing trend in spring, summer and autumn, while it kept constant in winter, similar to the value of $R_{NH4NO3}$ (red dotted line in Figure 4a). In addition, in spring, summer and autumn, the lowest values of the m/z 30/m/z 46 ratio, occurring at ~1 m, were approximate to the corresponding seasonal values of $R_{NH4NO3}$. It should be noted that the similar size distribution patterns of the m/z 30/m/z 46 ratio under the highest interferences (>15%) and lowest interferences (<5%) of $CH_2O_X^+$, indicated by the HR data of $PM_1$, for spring, summer and autumn (Figure S12) imply that the size distribution patterns of the m/z 30/m/z 46 ratio were not affected significantly by the interferences of $CH_2O_X^+$. We also used the size distributions of the m/z 30/m/z 46 ratio to separate the size distributions of inorganic and organic nitrates, as shown in Figure S13, and the results indicate that organic nitrates were relatively more concentrated at small sizes compared to inorganic nitrates."

2. SOA yield is dependent on organic aerosol mass. When using the SOA yields in literature to estimate SOA from VOC+NO3 (Table 3), please ensure that the yields used are from laboratory experiments that covered the relevant range of aerosol mass in the field campaigns. Please clarify this in the revised manuscript.

**REPLY:**

We have double checked the literature to make sure the chamber conditions to obtain the yields covered the range of aerosol mass loading in the spring campaign. The relevant clarification is added in the revised manuscript.

3. There are a number of grammatical mistakes / typos, e.g., line 303 should be "associated with", etc. Please correct them.

**REPLY:**

We have corrected them.

**A list of all relevant changes**

1. Line 197-202: Changed to "Unfortunately, due to the lack of HR-PToF data, our analyses used the UMR-PToF data of m/z 30 and 46, which might contain the interferences of $CH_2O_X^+$ (Fry et al., 2018). In our case, the time variations of contributions of $CH_2O^+$ in m/z 30 and $CH_2O_2^+$ in m/z 46 in the HR data of $PM_1$ for the four seasons are shown in Figure S10. For all the four seasons, the average contributions of $CH_2O_X^+$ in m/z 30 and 46 in the HR data of $PM_1$ were less than 10%, suggesting that the m/z 30/m/z 46 ratio could mostly represent the $NO^+/NO_2^+$ ratio. The average size distributions of m/z 30 and m/z 46 for the four seasons are shown in Figure S11".

2. Line 206-211: Added "It should be noted that the similar size distribution patterns of the m/z 30/m/z 46 ratio under the highest interferences (>15%) and lowest interferences (<5%) of $CH_2O_X^+$, indicated by the HR data of $PM_1$, for spring, summer and autumn (Figure S12) imply that the size distribution patterns of the m/z 30/m/z 46 ratio were not affected significantly by the interferences of $CH_2O_X^+$. We also used the size distributions of the m/z 30/m/z 46 ratio to separate the size distributions of inorganic and organic nitrates, as shown in Figure S13, and the results indicate that organic nitrates were relatively more concentrated at small sizes compared to inorganic nitrates".

3. Line 249-250: Added "where the chamber conditions to obtain the yields covered the range of aerosol mass loading in the spring campaign".

4. Line 305: Changed to "associated".

5. Line 383-387: Added "Fry, J. L., Brown, S. S., Middlebrook, A. M., Edwards, P. M., Campuzano-Jost, P., Day, D. A., Jimenez, J. L., Allen, H. M., Ryerson, T. B., Pollack, I., Graus, M., Warneke, C., de Gouw, J. A., Brock, C. A., Gilman, J., Lerner, B. M., Dubé, W. P., Liao, J., and Welti, A.: Secondary organic aerosol (SOA) yields from NO3 radical + isoprene based on nighttime aircraft power plant plume transects, Atmos. Chem. Phys., 18, 11663-11682, https://doi.org/10.5194/acp-18-11663-2018, 2018".

[revised manuscript text omitted]